# Mushroom body evolution demonstrates homology and divergence across Pancrustacea

**Nicholas James Strausfeld[1]\*, Gabriella Hanna Wolff[2]\*, Marcel Ethan Sayre[3]\***

[1]Department of Neuroscience, School of Mind, Brain and Behavior, University of Arizona, Tucson, United States; [2]Department of Biology, University of Washington, Seattle, United States; [3]Lund Vision Group, Department of Biology, Lund University, Lund, Sweden

**Abstract** Descriptions of crustacean brains have focused mainly on three highly derived lineages of malacostracans: the reptantian infraorders represented by spiny lobsters, lobsters, and crayfish. Those descriptions advocate the view that dome- or cap-like neuropils, referred to as 'hemiellipsoid bodies,' are the ground pattern organization of centers that are comparable to insect mushroom bodies in processing olfactory information. Here we challenge the doctrine that hemiellipsoid bodies are a derived trait of crustaceans, whereas mushroom bodies are a derived trait of hexapods. We demonstrate that mushroom bodies typify lineages that arose before Reptantia and exist in Reptantia thereby indicating that the mushroom body, not the hemiellipsoid body, provides the ground pattern for both crustaceans and hexapods. We show that evolved variations of the mushroom body ground pattern are, in some lineages, defined by extreme diminution or loss and, in others, by the incorporation of mushroom body circuits into lobeless centers. Such transformations are ascribed to modifications of the columnar organization of mushroom body lobes that, as shown in *Drosophila* and other hexapods, contain networks essential for learning and memory.

**\*For correspondence:**
flybrain@email.arizona.edu (NJS);
gabwolff@uw.edu (GHW);
marcel.sayre@biol.lu.se (MES)

**Competing interests:** The authors declare that no competing interests exist.

## Introduction

As demonstrated in *Drosophila melanogaster*, paired mushroom bodies in the insect brain play manifold roles in learning and memory (*Aso et al., 2014a*; *Aso et al., 2014b*; *Cognigni et al., 2018*). One diagnostic tool for identifying putative mushroom body homologues in other mandibulates is an antibody raised against the catalytic subunit of protein kinase A, encoded by the *Drosophila* gene DC0 (*Kalderon and Rubin, 1988*), and required for effective learning and memory (*Skoulakis et al., 1993*). The antibody, known as 'anti-DC0', selectively identifies the columnar neuropils of mushroom bodies of insects (*Farris and Strausfeld, 2003*) and of other arthropods with, until very recently, the notable exception of crustaceans (*Wolff and Strausfeld, 2015*). Indeed, numerous studies have disputed or expressed ambivalence about centers in comparable locations in the crustacean brain being mushroom body homologues (e.g., *Strausfeld et al., 1998*; *Fanenbruck et al., 2004*; *Fanenbruck and Harzsch, 2005*; *Farris, 2013*; *Sandeman et al., 2014*; *Krieger et al., 2015*; *Krieger et al., 2019*; *Harzsch and Krieger, 2018*; *Machon et al., 2019*; *Wittfoth et al., 2019*).

The discovery of mushroom bodies in mantis shrimps by *Wolff et al. (2017)* caused us to revise our view of mushroom bodies in crustaceans and to look for these centers in other lineages of this taxon. Here, we provide evidence that mushroom bodies generally occur in Crustacea, thereby further supporting Hexapoda + Crustacea as the established subphylum Pancrustacea (*Zrzavý and Štys, 1997*; *Regier and Shultz, 2001*).

Historically, identification of the insect mushroom body relied on just four neuroanatomical traits: an overall fungiform shape; a location exclusively in the brain's rostrolateral protocerebrum; an internal composition comprising hundreds to tens of thousands of intrinsic neurons originating from minute cell bodies clustered at the brain's rostral surface; and the possession of columnar lobes formed by the extended axon-like processes of intrinsic neurons. An often-included fifth character typical of almost all insects is a cap of neuropil called the calyx comprising the dendrites of intrinsic neurons, which receive inputs from the antennal lobes and other sensory neuropils. Even flightless Zygentoma (silverfish and firebrats) possess calyces (*Farris, 2005a*), as do Diplura and Collembola, two sister groups of Insecta (*Böhm et al., 2012*; *Kollmann et al., 2011*). However, Ephemeroptera (mayflies) and Odonata (dragonflies, darters) are calyxless; inputs to their mushroom bodies supply their columnar lobes directly (*Strausfeld et al., 2009*). Such distinctions demonstrate a general property of all well-defined protocerebral brain centers: retention of ancestral ground patterns despite evolved modifications such as losses of some components and elaborations of others. For the insect mushroom body these modifications include: variations of columnar lobe organization; expanded representations of sensory modalities; and loss, or hypertrophy, of the calyces (*Strausfeld et al., 2009*).

Despite such variations, the insect mushroom body has additional features that are always present. Foremost is that the columnar lobes are serially partitioned down their length such that inputs to the lobes are constrained to discrete synaptic domains (*Li and Strausfeld, 1997*; *Ito et al., 1998*; *Strausfeld, 2002*). These domains receive functionally defined inputs and supply functionally defined mushroom body output neurons (MBONs; *Aso et al., 2014a*) that extend to circumscribed regions of the medial and rostral protocerebrum. There, they contribute to further higher processing and have arrangements comparable to those relating the mammalian hippocampus to cortical areas (*Bienkowski et al., 2018*). Certain outputs from the columns also project recurrently to more distal levels of the mushroom body itself (*Gronenberg, 1987*; *Zwaka et al., 2018*). These arrangements are defined by neurons usually expressing GABA (*Liu and Davis, 2009*). The same organization typifies the columnar lobes of the stomatopod mushroom bodies where input and output neurons are arranged along the length of the lobes (*Wolff et al., 2017*).

Also apparent across insect species is that subsets of intrinsic neurons parse the mushroom body lobes into longitudinal subunits that are differentiated from each other with regard to their clonal lineages (*Yang et al., 1995*; *Crittenden et al., 1998*; *Tanaka et al., 2008*), as well as by their aminergic and peptidergic identities (*Sjöholm et al., 2006*; *Strausfeld et al., 2000*; *Strausfeld et al., 2003*). Comparable longitudinal divisions of the stomatopod mushroom body are represented by four columnar lobes that extend together approximately in parallel (*Wolff et al., 2017*). In *Drosophila* and other insects these characteristic attributes have been shown to be critical in supporting functions relating to the modality, valence, stored memories and behavioral relevance of information computed by the mushroom body (*Li and Strausfeld, 1999*; *Aso et al., 2014a*; *Aso et al., 2014b*; *Owald et al., 2015*; *Takemura et al., 2017*; *Hattori et al., 2017*; *Cognigni et al., 2018*).

In total, thirteen characters have been identified that define both the insect (*Drosophila*) and mantis shrimp (Stomatopod) mushroom bodies (see *Wolff et al., 2017*). We show here that crown eumalacostracan species belonging to lineages originating early in evolutionary history share the same expanded set of diagnostic characters that define insect mushroom bodies, including the partitioning of columns into discrete circuit-defined domains. These can occur not only as segment-like partitions of the columnar lobes but also as tuberous outswellings, as they do in basal insects belonging to Zygentoma and Pterygota (*Farris, 2005a*; *Farris, 2005b*; *Strausfeld et al., 2009*).

Because stomatopods are unique amongst crustaceans in possessing elaborated optic lobes that serve a multispectral color and polarization photoreceptor system (*Thoen et al., 2017*; *Thoen et al., 2018*), it could be argued that mushroom bodies in the mantis shrimp are unique apomorphies that have evolved specifically to serve those modalities. However, Stomatopoda are an outgroup of Eucarida (Euphausiacea + Decapoda) and the status of mushroom bodies as the ancestral ground pattern is supported by corresponding centers in the lateral protocerebrum of later evolving eumalacostracan lineages. For example, *Lebbeus groenlandicus*, a member of the caridid family Thoridae, has been shown to possess paired mushroom bodies each comprising a layered calyx supplying intrinsic neuron processes to columnar lobes (*Sayre and Strausfeld, 2019*). Here we describe neuroanatomical characters defining mushroom bodies also in cleaner shrimps (Stenopodidae) and several groups of carideans.

The recognition that mushroom bodies occur in crustaceans is not new. In his 1882 description of stomatopod brains, the Italian neuroanatomist Giuseppe Bellonci identified domed neuropils (corpo emielissoidale), from which extend columnar neuropils (corpo allungato). Bellonci explicitly homologized these with, respectively, the insect mushroom body calyx and its columnar lobes (*Bellonci, 1882*). Hanström adopted this terminology for his studies of Reptantia (*Hanström, 1925*; *Hanström, 1931*), a relatively recent malacostracan lineage that includes crayfish and lobsters. Hanström used the terms 'hemiellipsoid body' or corpora pedunculata to denote centers lacking columnar lobes he considered to be homologues of the mushroom body (*Hanström, 1932*). However, over the last four decades numerous studies on Reptantia have insisted that hemiellipsoid bodies are apomorphic – genealogically distinct from mushroom bodies – and that they represent the ancestral ground pattern of crustacean learning and memory centers (*Kenning et al., 2013*; *Sandeman et al., 2014*; *Machon et al., 2019*; *Krieger et al., 2019*). Apart from a fundamental misunderstanding of the original meaning of the term 'hemiellipsoid body,' that viewpoint is incorrect and is in conflict with the demonstration here, and in two recent studies, that mushroom bodies hallmark eumalacostracan lineages that diverged earlier than Reptantia (*Wolff et al., 2017*; *Sayre and Strausfeld, 2019*). Studies of Reptantia also show that at least one anomuran group has mushroom bodies equipped with columnar lobes (*Strausfeld and Sayre, 2020*). Comparisons across Eumalacostraca described here indicate that mushroom bodies are indeed ubiquitous across Crustacea. However, in contrast to mushroom bodies in insects, evolved modifications of the mushroom body ground pattern in crustaceans have resulted in highly divergent morphologies including, both within Caridea and in certain lineages of Reptantia, centers lacking defined lobes.

## Results

In the following, descriptions of mushroom bodies and their evolved derivatives are organized by lineage from the oldest to most recent as shown in *Figure 1A*. The exception is Leptostraca, the sister group of Eumalacostraca, which is considered in the Discussion. Each of the following descriptions is preceded by a brief *Background* of the species considered followed by *Observations*. To provide context we first provide an overview of mushroom body organization as exemplified by the stomatopod brain (*Figure 2*).

### The evolutionary timeline

Species considered here belong to malacostracan lineages whose divergence times are known from fossil-calibrated molecular data (*Wolfe et al., 2019*), and which are estimated to have originated between the mid-to-late Ordovician and the Carboniferous (*Figure 1A*). The following descriptions include only a small number of the species belonging to eucarid lineages used for molecular phylogenomic reconstruction (see *Schwentner et al., 2018*; *Wolfe et al., 2019*). Here we also refer to analyses that focus on mantis shrimps (*Van Der Wal et al., 2017*), as well as decapods, including stenopids (cleaner shrimps), alpheids and carideans ('visored shrimps', 'pistol shrimps': *Anker et al., 2006*; *Anker and Baeza, 2012*; *Bracken et al., 2010*; *Davis et al., 2018*), brachyurans ('true crabs': *Tsang et al., 2014*), thalassinids ('ghost shrimps': *Tsang et al., 2008*), anomurans including 'hermit crabs' (*Bracken-Grissom et al., 2013*; *Chablais et al., 2011*), and various clades that colloquially are referred to as lobsters (see *Shen et al., 2013*; *Bracken-Grissom et al., 2014*). *Figure 1B* provides a summary of anti-DC0-immunoreactive territories described as homologues of the mushroom body ground pattern, detailed descriptions of which follow.

### General organization of the lateral protocerebrum

The principal organization of the olfactory pathway of crown Crustacea up to the level of the lateral protocerebrum is reminiscent of that in Hexapoda although claims of homology (*Schachtner et al., 2005*; *Harzsch and Krieger, 2018*) may be insecure (see Discussion and *Table 1*). With the exception of Cephalocarida (*Stegner and Richter, 2011*), axons of relay neurons from olfactory centers in the deutocerebrum ascend rostrally as two prominent mirror-symmetric fascicles, called the olfactory globular tracts (OGT; see *Figure 2—figure supplement 1*). These axonal pathways are massive, often comprising many thousands of axons. Left and right OGTs converge in the mid-protocerebrum, just dorsal to the central body where most if not all their axons bifurcate sending a tributary into both lateral protocerebra (*Figure 2A*; *Figure 2—figure supplement 1*). In Eumalacostraca, the

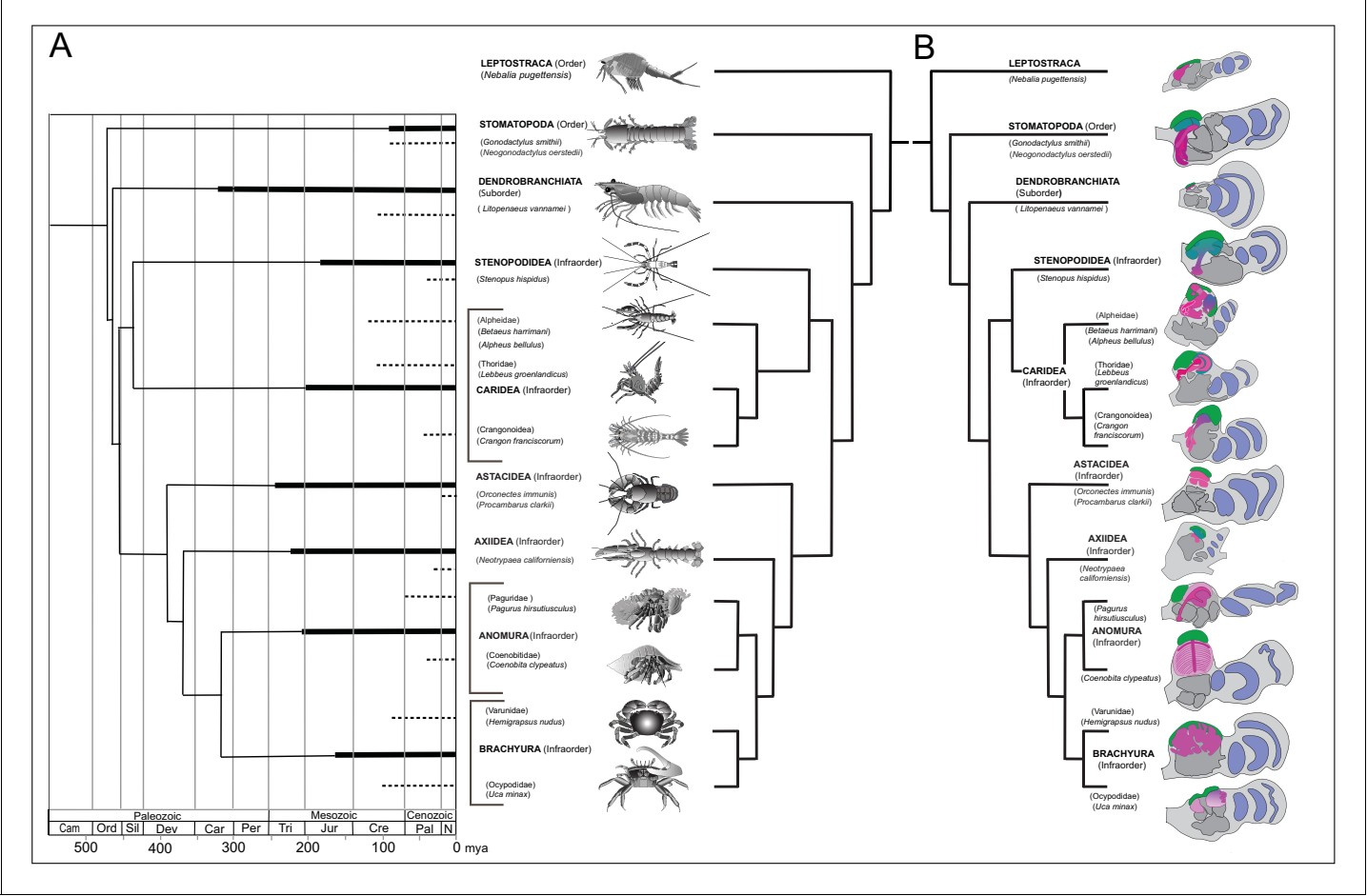

**Figure 1.** Time lines, phylogeny and protocerebral morphology of malacostracan lineage representatives. Time lines and lineage relationships are based on the molecular phylogeny of *Wolfe et al. (2019)*. (**A**) Geological time scale shown as millions of years ago (mya). Solid lines indicate estimated occurrence of lineages sampled for this study; dashed lines indicate estimated age of representative taxa (see citations in text). Images depict species used for this study. (**B**) Schematics showing proportions of anti-DC0-immunoreactive centers (shades of magenta) in the right lateral protocerebrum of species described in this account. Rostral is up, distal to the right. Nested optic lobe neuropils shown blue; rostrally disposed globuli cell clusters, green; generalized neuropil domains of the lateral protocerebrum, mid-gray.

lateral protocerebral neuropils, which include the nested optic lobe centers, are usually resident within the enlarged volume of the eyestalk, immediately beneath the compound retina. However, in land hermit crabs the lateral protocerebral neuropils are located at the base of the eyestalks. Other exceptions are in lineages that lack eyestalks (species of Alpheidae, pistol shrimps, hooded shrimps, and Thalassinidae [ghost shrimps]), or lack compound eyes (e.g., Copepoda and Remipedia), where the lateral protocerebra are incorporated into, or bulge outwards from, the midbrain. Mushroom bodies are situated in the lateral protocerebrum, as are their morphological variants, which have corresponding topographical relationships with other neuropils (*Figure 2—figure supplement 1*).

## The stomatopod mushroom body (*Figure 2*)

### Background

To introduce the crustacean mushroom body, we briefly recap and expand our findings from Stomatopoda (*Figure 2*; see also, *Wolff et al., 2017*). Stomatopoda is sister to all other Eumalacostraca. Its lineage has been claimed as extending back to the Devonian, represented by fossil Tyrannophontidae (*Hof, 1998*), but the relationship of this fossil to extant stomatopod morphology is highly problematic. Unambiguous fossil Stomatopoda are Cretaceous (*Wolfe et al., 2016*), and their origin by molecular clock data is Triassic (*Van Der Wal et al., 2017*).

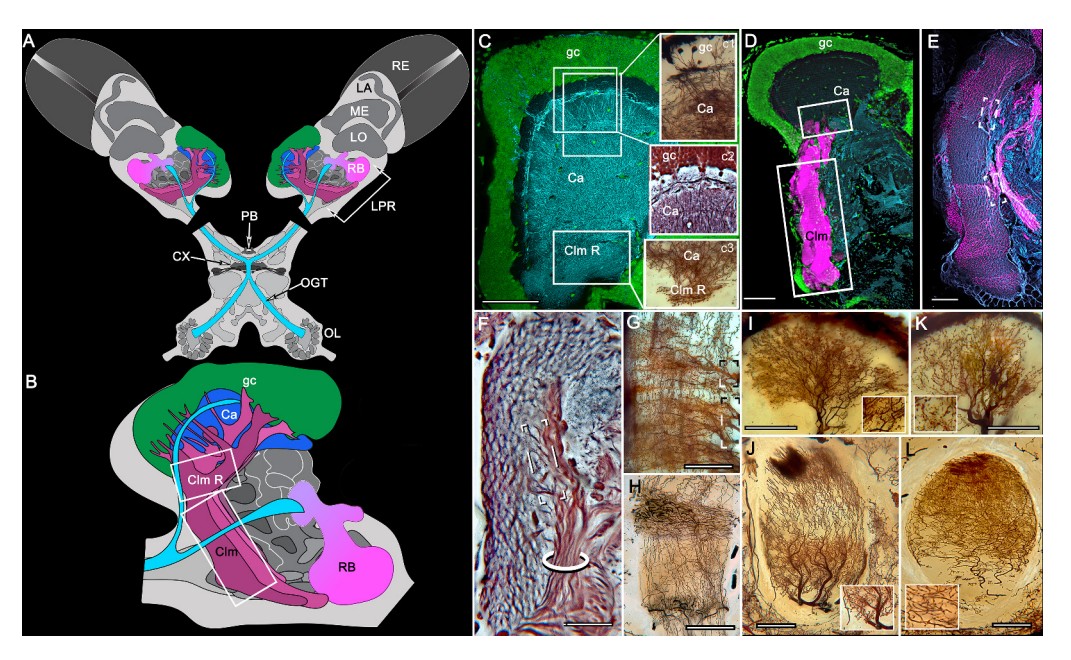

**Figure 2.** Disposition and cardinal features of the stomatopod mushroom body. (A) Schematic showing the relative proportions of the lateral protocerebra (LPR) and medial protocerebrum. The latter is defined by the central body (CX), protocerebral bridge (PB), and associated neuropils located anterior to the deutocerebrum, which is denoted by its olfactory lobes (OL). Axons of olfactory relay neurons leave the olfactory lobes to provide the olfactory globular tract (OGT, cyan) that terminates in the mushroom body calyces and more rostral neuropils of the lateral protocerebrum (LPR). (B) Enlargement of the stomatopod mushroom body showing the globuli cell layer (gc) supplying intrinsic neurons that branch in the calyces (Ca). Their axon-like fibers converge at the columnar root (Clm R) and extend through the lengths of the columnar lobes (Clm). In this species (*Neogonodactylus oerstedii*), four lobes (dark magenta) extend in parallel. A second prominent neuropil is the reniform body (RB), which like the columnar lobes is immunoreactive to anti-DC0. Here and in other species, the RB resides between the mushroom body and optic lobes beneath the retina (RE; lamina, LA; medulla, ME; lobula, LO). (C-L) Histological features that define the mushroom body. Globuli cells (gc; green Syto13) provide thousands of overlapping dendritic trees that populate the calyx (Ca; cyan: anti-α-tubulin). Axons of intrinsic cells converge deep in the calyx to provide the roots of the columnar lobes (Clm R). The boxed areas in C refer to panels (c1-3). Panel c1 shows Golgi-impregnated globuli cells (gc), with neurites extending into the dense layered meshwork of intrinsic neuron dendrites that comprise the calyx; Bodian staining (c2) resolves the packed globuli cells overlying their stratified dendrites in calycal layers. Golgi impregnation (c3) shows axon-like processes converging at the base of the Ca where they form the root of the columnar lobe (Clm R). (D) Confocal laser scan showing anti-DC0 labelling along the length of a columnar lobe (Clm; magenta, anti-DC0; cyan: anti-α-tubulin labelled neuropil). The small rectangle denotes the origin of the column from the calyx, as shown in panel B. The larger rectangle corresponds to comparable lengths of columnar neuropil in panels B, E and F. (E) Anti-tyrosine hydroxylase (magenta) and anti-α-tubulin (cyan) immunolabelling reveals discrete synaptic domains of mushroom body output neurons (MBON) along the length of a column. Their exit points are indicated by open boxes. (F) Bodian-stained section showing braid-like bundles of intrinsic neuron parallel fibers. The open box indicates the exit points of MBONs, the axons of which converge to form a prominent bundle (ringed) destined for volumes of the rostral midbrain protocerebrum (not shown). (G) Golgi-impregnated afferent and efferent processes enter at discrete domains (open boxes) along the stomatopod's mushroom body column. (H) For comparison: organization of homologous neurons in the mushroom body column of the cockroach *Periplanata americana* . (I, J) Cross sections of mushroom body column in a stomatopod (panel I) and cockroach (panel J) demonstrate spine-like specializations (insets) of dendrites typifying MBONs. (K, L) Cross sections of afferent terminals in a stomatopod columnar lobe (panel K) and cockroach (panel L). Insets show corresponding beaded specializations. Scale bars in A, 50 μm; B, 100 μm; C-G, 50 μm; H, 100 μm; I, 50 μm; J, 100 μm.

The online version of this article includes the following figure supplement(s) for figure 2:

**Figure supplement 1.** Ground pattern of the pancrustacean olfactory pathway and its lateral protocerebral termini.

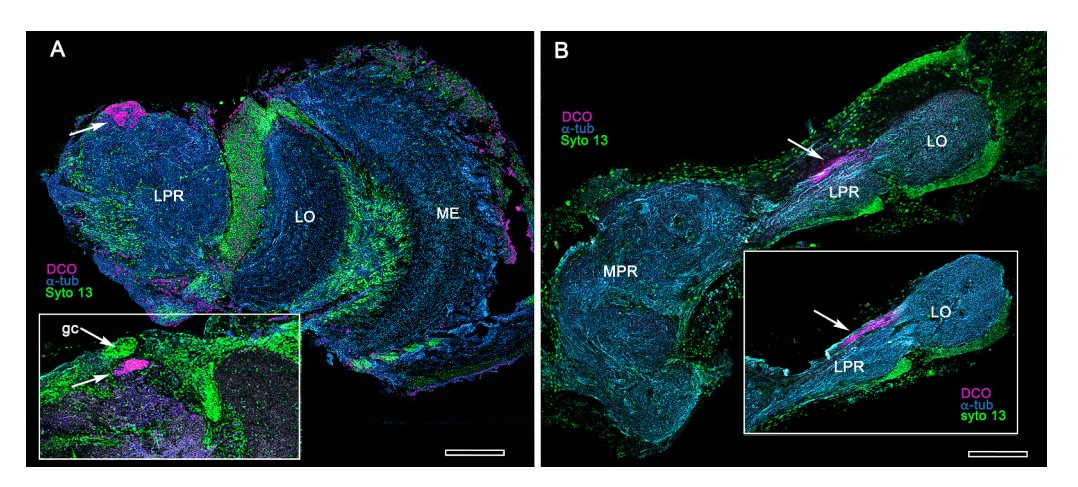

**Figure 3.** Extreme reduction of DC0-immunoreactive neuropils in decapods and isopods. (**A**) The pelagic white leg shrimp *Penaeus vannamei*, showing substantial lateral protocerebral neuropils (LPR) and optic neuropils (ME, medulla; LO, lobula), but a minute anti-DC0-immunoreactive center, the location of which corresponds to that of the ancestral mushroom body of Stomatopoda and Caridea, and the derived hemiellipsoid body of Reptantia. Inset: Another example showing also the clustered globuli cells (gc) overlying the anti-DC0-immunoreactive center. (**B**) Preparations from two individual isopods *Ligia pallasii*, showing the medial protocerebrum (MPR) connected to a greatly reduced lateral protocerebrum (LPR). A narrow layer of anti-DC0-immunoreactive neuropil resides at a position corresponding to that occupied by the mushroom body of a stomatopod or the hemiellipsoid body of a reptantian. Magenta, anti-DC0; cyan, α-tubulin; green, Syto13. Scale bars,100 μm.

## Observations

In Stomatopoda, the mushroom body's intrinsic neurons originate from adjoining clusters of basophilic cell bodies (globuli cells). These contribute dendrites to a large calyx from which arise parallel columnar extensions (*Figure 2A–C*). In total, four columns impart a quadripartite organization corresponding to that of an insect mushroom body (*Wolff et al., 2017*; *Ito et al., 1997*). Further correspondences are the arrangements of intrinsic neurons in the stomatopod calyces, their parallel axon-like extensions into the lobes, and the division of the lobes into discrete synaptic domains. Notably, intrinsic neuron dendrites define at least three discrete layers through the calyces (*Figure 2C*, inset c1, c2). Their axon-like prolongations converge at the base of the calyces to form columnar lobes (*Figure 2C*, inset c3). In Stomatopoda and, as will be described, in other species, the lobes are intensely labelled by antibodies raised against DC0 (*Figure 2D*), the *Drosophila* orthologue of vertebrate PKA-Cα, which in vertebrates and invertebrates plays a crucial role in synaptic facilitation (*Burrell and Sahley, 2001*; *Abel and Nguyen, 2008*). Numerous other characters defining the stomatopod mushroom body correspond to those resolved in a basal dicondylic insect, exemplified by the cockroach *Periplaneta americana,* as well as in more recent groups such as Drosophilidae. For example, synaptic domains partitioning the columnar lobes are defined by modulatory and peptidergic efferent and afferent processes (*Figure 2E*). Large numbers of these supply the braided organization of parallel fibers that comprise the lobes (*Figure 2F*). The processes of intrinsic neurons form characteristic orthogonal 'Hebbian' networks as they do in insects (*Figure 2G,H*). And although the cross-sectional profile of the mushroom body column may vary across species – it is oval in *Periplaneta* for example, but more discoid in the stomatopod – the spinous and varicose attributes of, respectively, efferent (Figure 12I,K) and afferent neurons (Figure 12J,L) are identical in stomatopods and insects.

## Reduced protocerebral centers in Dendrobranchiata (*Figure 3A*)
### Background
Dendrobranchiata is the second oldest eumalacostracan lineage, today represented by peripatetic/pelagic penaeid shrimps. Using antibodies raised against α-tubulin, FMRFamide, and allatostatin on *Penaeus vannamei*, *Meth et al. (2017)* were unable to delineate a target neuropil supplied by the

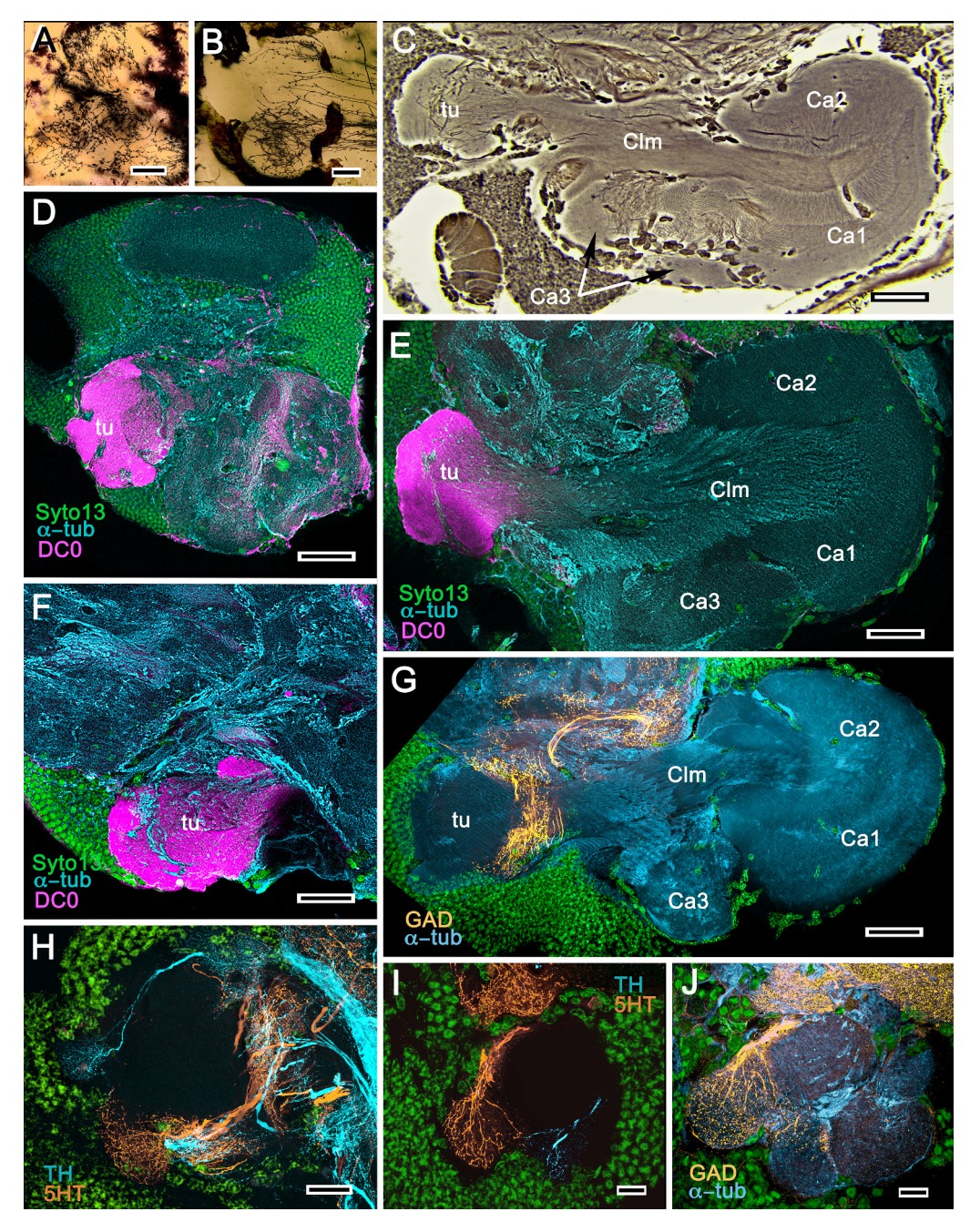

**Figure 4.** Mushroom bodies of *Stenopus hispidus*: DC0 and neuromodulatory delineation of tubercular domains. (**A,B**) Golgi-impregnated neurites arborizing in the mushroom body columns' tubercles. (**C**) Bodian silver-stained overview of the stenopid mushroom body. Axon-like extensions from three distinct calycal regions (Ca1, Ca2, Ca3) provide the column (Clm) that terminates in discrete tubercular domains (tu). (**D–F**) Immunolabelled sections showing regions containing elevated levels of anti-DC0 (magenta). Anti-DC0 immunoreactivity dominates the entirety of the column's tubercular domain (cyan, α-tubulin; green, Syto13). (**G**) Calycal domains and the column show sparse, if any, anti-GAD immunoreactivity (yellow), with the exception of processes that innervate a zone spanning the width of the column immediately distal to its tubercles (tu, cyan: anti-α-tubulin). Neuron cell bodies are shown stained green with Syto13. (**H,I**) Anti-serotonin (5HT, orange) and anti-tyrosine hydroxylase (TH, cyan) immunoreactive fibers similarly occupy discrete territories in the column's tubercle. Neural and glial cell bodies are shown stained green with Syto13. (**J**) Tubercular domains showing discrete territories, each a subset of intrinsic

*Figure 4 continued on next page*

*Figure 4 continued*
neuron processes, occupied by anti-GAD-immunoreactive processes (yellow; cyan, α-tubulin; green, Syto13). Scale bars in A, B, 20 μm; C, 50 μm; D-H, 50 μm; I, J, 20 μm.

olfactory globular tract (OGT). *Sullivan and Beltz (2004)*, using dye tracing into the OGT of *Penaeus duorarum,* identified a small twinned neuropil situated extremely rostral in the lateral protocerebrum. The neuropil receives collaterals from the OGT, the axons of which terminate mainly in lateral protocerebral neuropil beneath.

## Observations
Immunostaining the lateral protocerebra of *Penaeus vannamei* with anti-DC0 (*Figure 3A*) resolves a small intensely immunoreactive four-component center, at the location identified by *Sullivan and Beltz (2004)*. A similarly reduced center typifies some isopods (*Figure 3B*). There is a notable absence of a columnar lobe in the *Penaeus* center and in the corresponding neuropil of isopods.

## Elaborated mushroom bodies in Stenopodidea (*Figures 4* and *5*)
### Background
Mitochondrial genomics resolves the infraorder Stenopodidea as the sister taxon of Caridea (*Shen et al., 2013*). Its molecular clock age is Jurassic with a Cretaceous fossil calibration (*Wolfe et al., 2019*). The following description is restricted to the coral banded cleaner shrimp *Stenopus hispidus*, one of the few well-documented cleaner shrimps that inhabit tropical coral reefs and groom fish of external parasites or detritus (*Vaughan et al., 2017*). These shrimps are well known for their knowledge of place, occupying a specific territory – their cleaning station – to which they return daily and which is visited by favored fish (*Limbaugh et al., 1961*). Long-term memory in *S. hispidus* is suggested by observation of individuals pair bonding and recognizing a partner even after an absence of several days (*Johnson, 1977*). Allocentric recognition of place and partner recognition are mediated by chemoreception (*Esaka et al., 2016*).

### Observations
We confirm here the original description by *Sullivan and Beltz (2004)* of the *S. hispidus* lateral protocerebral center that they referred to as a hemiellipsoid body divided into three contiguous domains that receive layered OGT terminals. A neuropil named by Sullivan and Beltz as the 'lateral protocerebral complex' and described as an entirely separate neuropil, as it is by *Krieger et al. (2019)*, is clearly a columnar lobe originating from these domains and terminating as a set of tubercular swellings, (*Figure 4A–C*). The swellings are strongly immunoreactive to anti-DC0 (*Figure 4D–F*). Reduced silver staining (*Figure 4C*) and immunohistology (*Figure 4E*) demonstrate that the three domains (Ca1-Ca3) assume the configuration of a layered mushroom body calyx. Intrinsic neurons in each domain (*Figure 5A,B,D*) contribute many hundreds of axon-like fibers that converge into a common root, from which arises the compound columnar lobe terminating as three adjacent tubercular domains, corresponding to the three calyces (*Figures 4D–F* and *5A,B*). Parallel fibers branch in these tubercles where they provide pre- and postsynaptic specializations (*Figure 4A,B*). As remarked above, anti-DC0 is expressed at high levels in the tubercles and at lower levels just distally in the columnar lobe (*Figure 4D–F*). And, as shown by immunolabelling with antibodies against GAD, 5HT and TH, the tubercles are further divided into smaller territories such that the entire structure assumes the serial domain arrangement characterizing the mushroom body ground pattern (*Figure 4G–J*).

The calyces correspond to Sullivan and Beltz's 'hemiellipsoid body.' They are composed of many thousands of intrinsic neuron dendrites. Their axon-like extensions, resolved by Golgi impregnations (*Figure 5A*), converge predominantly beneath Ca2 in the columnar root but also recruit from both Ca1 and Ca3 through an elaborate system of interweaving processes (upper and lower insets, *Figure 5A*). Groups of parallel fibers segregate before invading different tubercles, each denoting a discrete integrative domain.

As in *Lebbeus* and Stomatopoda, the calyces are profusely invaded by branches of TH-containing neurons (*Figure 5B*). These are arranged through three successive calycal layers (*Figure 5C–E*), each

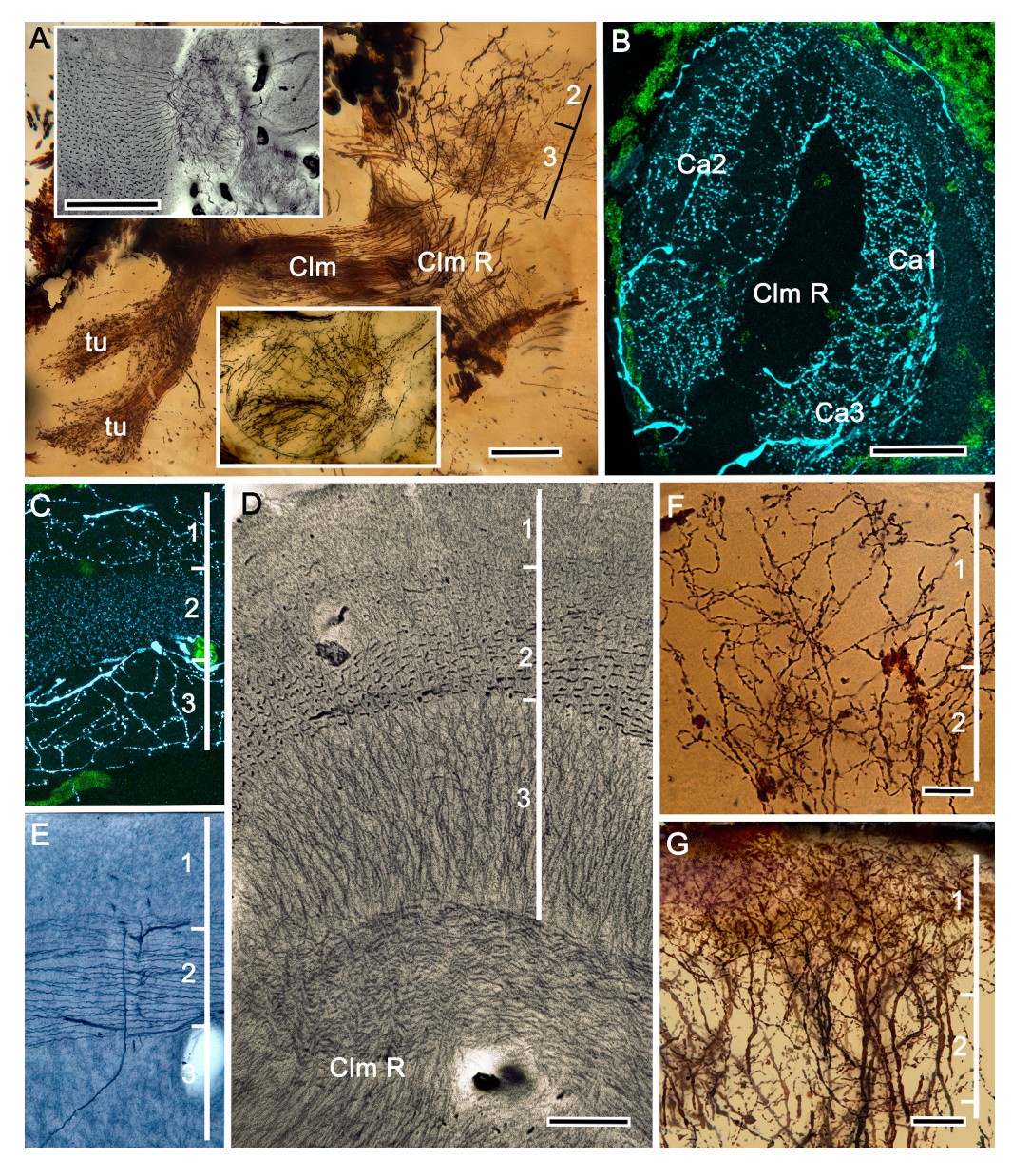

**Figure 5.** The stenopid mushroom body: cytoarchitecture of the calyx. (**A**) Mushroom body intrinsic neurons extend their axon-like processes from the calyx into the column (Clm) terminating in its tubercles (tu). Levels 2, 3 indicate the two inner layers of the calyx. Upper inset (Bodian reduced silver) and lower inset (Golgi- impregnated processes) show axon-like extensions from intrinsic neurons undergoing elaborate sorting to enter the columnar lobe; thus comprising three fused lobes, one from each calyx. (**B**) Anti-tyrosine hydroxylase immunolabelling (TH, cyan; cell bodies, green) reveals the supply by anti-TH-immunoreactive processes to all three calyces surrounding the root of the mushroom body column (Clm R). (**C**) Enlargement of the calyx showing anti-TH-positive neuronal arborizations in calycal layers 1 and 3, while anti-TH labelling in layer 2 shows very fine, yet densely populated neuron varicosities. (**D**) Bodian stain showing the stratified stenopid calyx. Layer 1 contains the apical dendrites of mushroom body intrinsic neurons, as well the terminals of afferent neurons from other lateral protocerebral centers. Terminals of the olfactory globular tract (OGT) arborize in layer 2. Layer 3 is occupied by the extensions of the inner dendrites of intrinsic neurons. Intrinsic neurons send their axon-like extensions into the root of the columnar lobe (Clm R; also see **A**), top inset) after extending laterally beneath layer 3. (**E**) Bodian stain of layer 2, which is further defined by numerous parallel projecting terminals from the olfactory lobes carried by the OGT. (**F, G**) Golgi impregnations of slender, varicose afferent neurons (**F**) and apical dendrites of intrinsic neurons (**G**) in layer 1. Scale bars In A, 20 μm; B, 50 μm; C-E, 20 μm; F, 100 μm; G, 50 μm.

of which is defined by its elaborate cytoarchitecture reflecting specific neuronal components, as described from Stomatopoda and the alpheoid shrimp *Lebbeus* (*Wolff et al., 2017*; *Sayre and Strausfeld, 2019*). Layer 1 of each of the three calyces (Ca1-Ca3) contains ascending processes of slender varicose afferents and the apical dendrites of intrinsic neurons (*Figure 5F,G*). Climbing fibers enter the calyces from other lateral protocerebral centers. Layer 2 (*Figure 5D,E*) is denoted by the numerous parallel projecting terminals originating from the olfactory globular tract. Layer 3 (*Figure 5D*) contains the massively arborizing inner dendrites of intrinsic neurons, some of which are shown at the corresponding level in *Figure 5A*, upper right. Processes of intrinsic neurons sweep laterally beneath layer 3 to enter the root of the mushroom body's columnar lobe (Clm R; *Figure 5A, D*).

## Caridean mushroom bodies (and reniform body) (*Figures 6–11*)
### Background
Caridea splits from its sister group Procaridea in the Carboniferous (*Bracken et al., 2010*), with most caridean crown group diversification occurring in the Jurassic (*Wolfe et al., 2019*). Caridea comprises the second largest decapod taxon, with two-thirds of its species occupying marine habitats. Here, we consider marine species belonging to the families Alpheidae and Thoridae (members of the superfamily Alpheoidea), and intertidal species of the family Crangonidae. These families originated approximately 220, 160 and 70 mya, respectively (*Davis et al., 2018*). Alpheidae is a diverse family that contains snapping shrimps, commonly referred to as pistol shrimps, which include the only known crustaceans to have evolved eusociality (*Duffy et al., 2000*), and visored shrimps closely related to the genus *Alpheus* (*Anker et al., 2006*). Pistol shrimps are represented here by the non-social species *Alpheus bellulus*, which, like other snapping shrimps, possesses asymmetric claws (chelae). The visored shrimp is *Betaeus harrimani*. Like pistol shrimps, *B. harrimani* is an active predator, although not employing concussion, and is also facultatively commensal, sharing burrows of the ghost shrimp *Neotrypaea californiensis* (*Anker and Baeza, 2012*). Unlike the genus *Alpheus*, *Betaeus* has symmetric claws, reflecting the more basal position of this genus within Alpheidae (*Anker et al., 2006*).

### Observations
Comparisons of the brains of *A. bellulus* and *B. harrimani* revealed almost no discernible differences other than that the eyes and visual neuropils of *B. harrimani* are smaller than those of *A. bellulus*. *Figures 6* and *7* therefore illustrate relevant features of the mushroom body using both species. Immunolabelling with anti-DC0 reveals an elaborate system of neuropils expressing high levels of the antigen, demonstrating the relationship of those areas predominantly to the globuli cell cluster medially and to the protocerebrum rostrally (*Figure 6A,B*). Notably, alpheid species lack eyestalks and possess very small compound eyes. The nested optic centers (lamina, medulla and lobula) are correspondingly reduced, and the enormous mushroom bodies occupy much of the rostral and lateral protocerebrum's volume (*Figure 6A,B*). The calyx is elaborate, showing through its depth high affinity to DC0 (*Figure 6C*). As in Stomatopoda and Stenopodidea, the calyx has three layers, each of which is defined by the dendritic disposition of its intrinsic cells (*Figure 6D–F*). Intrinsic cells extend their axon-like processes in parallel bundles for a short distance before forming elaborate networks in the columnar lobes (*Figure 6D,E*), which extend dorso-ventrally as convoluted volumes comprising tubercular domains (*Figure 6G*). These volumes are heavily invested by anti-TH-, anti-5HT- and anti-GAD-positive arborizations (*Figure 6H,I*).

As in Stomatopoda and varunid Brachyura (shore crabs), the alpheid protocerebrum has a prominent DC0-positive reniform body, the components of which correspond to those described for *Neogonodactylus oerstedii* and *Hemigrapsus nudus* (*Thoen et al., 2019*). This center consists of four discrete neuropils linked by a branching axonal tract (pedestal) that extends axons from a glomerular initial zone to three separate zones (lateral, distal and proximal) each with its own characteristic immunomorphology (*Figure 7A–E*). Abutting the mushroom body ventrally, these territories are similar to, but spatially distinct from, a columnar extension from the mushroom body calyx that, like the main columnar ensemble shown in *Figure 6B,G*, consists of discrete tubercles, each denoted by its affinity to antisera against GAD, DC0, TH and 5HT (*Figure 7F–I*).

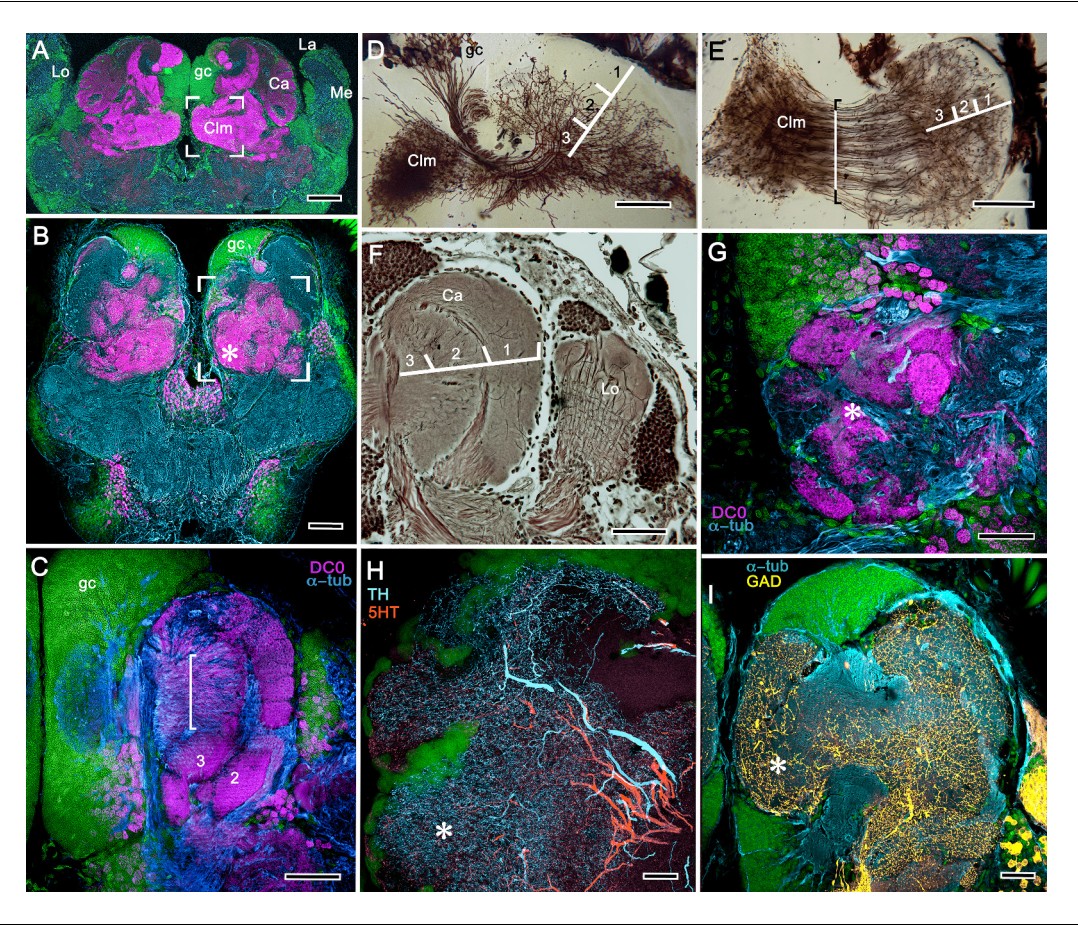

**Figure 6.** Neuroarchitecture of the alpheid mushroom body. **(A)** Anti-DC0-immunoreactivity (magenta) in the brain of *Alpheus bellulus*. The mushroom bodies (Ca calyx; Clm columnar lobe) are flanked on both sides of the brain by the optic lobes (cyan, anti-α-tubulin; green, Syto13; La, lamina; Me, medulla; Lo, lobula). **(B)** Anti-DC0 (magenta) immunolabelling in the brain of *Betaeus harrimani*. Asterisk marks a columnar lobe ending in tubercles (cyan, anti-α-tubulin; green, Syto13). **(C)** The mushroom body calyx of *A. bellulus*. Throughout its depth, and in each layer (layers 2 and 3 are shown here), the calyx shows a high affinity for antibodies raised against DC0. Striations (bracket) correspond to bundles of intrinsic neuron processes supplying the columnar lobe, correspondingly bracketed in panel E. **(D,E)** Golgi impregnations of mushroom body intrinsic cell clusters, originating from globuli cells (gc) and giving rise to dendrites in the calyces (Ca, layers numbered 1–3) and column (Clm). The distinct calycal layers are most clearly revealed in Bodian-stained sections **(F)**. **(G)** Anti-DC0 is expressed in distinct territories within the *A. bellulus* tubercles. **(H, I)** Aminergic processes in the lateral protocerebrum of *B. harrimani*. Asterisks in G, H, I denote corresponding regions. **(H)** Anti-5HT (orange) and anti-TH (cyan) show these neural arborizations invading most regions of the lateral protocerebrum coincident with volumes denoted by anti-DC0-labelled mushroom body-associated structures. **(I)** Anti-GAD (yellow) immunoreactivity shows an expression pattern throughout the anti-DC0-positive domains shown in B (cyan, anti-α-tubulin). Scale bars in A (*A. bellulus*), 100 µm; B (*B. harrimani*), 100 µm; C-F (*A. bellulus*), 100 µm; G (*A. bellulus*), 40 µm; H (*B. harrimani*), 20 µm; I (*B. harrimani*) 50 µm.

We next consider Thoridae, a separate lineage of the superfamily Alpheoidea that includes the genera *Lebbeus* and *Spirontocaris*.

## Backgound

*Lebbeus groenlandicus* and *Spirontocaris lamellicornis* are exotically patterned shrimps and protected from predation by sharp spiny protuberances from their exoskeleton. They are solitary, highly active, and territorial, living on reefs or rocks. The two species look similar; both are patterned into irregular white and orange-pink bands, a feature suggesting self-advertisement or aposomatic

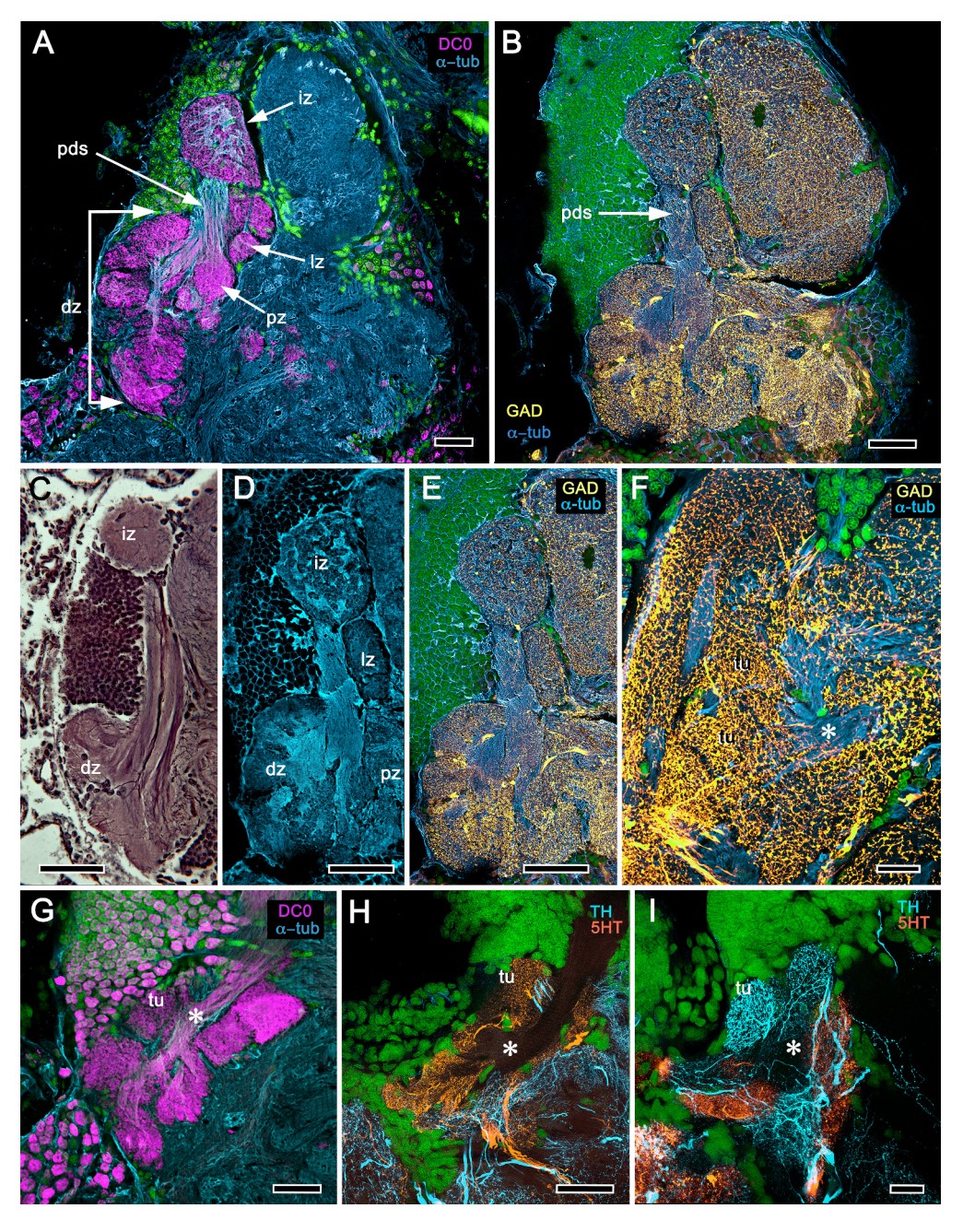

**Figure 7.** Reniform body (A–E) and mushroom body lobes (F–I) in alpheid shrimps. (A-E) Neuroarchitecture of the *Alpheus bellulus* reniform body (RB; cyan, anti-α-tubulin; green, Syto13). (A) Components of the alpheid reniform body are clearly distinguished by their high affinity for anti-DC0 (magenta). (A) pedestal-like (pds) bundle of neurites gives rise to four distinct zones: the initial zone (iz), the proximal zone (pz), the distal zone (dz) and the lateral zone (lz). (B,E) Anti-GAD immunolabelling (yellow) reveals putative inhibitory processes occupying major zones of the reniform body, other than the pedestal. (C) Bodian silver-stained *Betaeus harrimani* and (D) anti-α-tubulin-labelled *A. bellulus* sections reveal the nearly identical layout of the reniform body across the two alpheid species. (F–I) Anti-DC0 and aminergic innervation of discrete domains within tubercles branching off the mushroom body's columnar lobe. Asterisks in panels F–I indicate the MB column with its processes defasciculate into discrete tubercular domains. One corresponding tubercle (tu) is indicated in panels G-I. (F) Innervation of tubercular domains by anti-GAD-immunoreactive processes (yellow). (G) Anti-DC0 labelling expressed within tubercles. (H, I) Tubercular domains delineated by anti-TH- and anti-5HT-immunoreactive fibers. In all panels

*Figure 7 continued on next page*

*Figure 7 continued*

green indicates Syto13-stained neuronal cell bodies. Scale bars in A-C,E, 50 µm (*B. harrimani*); D,G-I, 50 µm; F, 10 µm (*A. bellulus*).

## Observations

*Figures 8*, *9* and *10A* expand a recent study of *Lebbeus* (*Sayre and Strausfeld, 2019*), resolving two completely distinct calyces that share a common organization into three concentric layers. These correspond to the dense arrangements of intrinsic neurons (inset to *Figure 8A*). Defined by their immunoreactivity to DC0 and affinity to a palette of antibodies (*Figure 8A,B,D*), the layers in the larger calyx (Ca1, *Figure 8A*) demonstrate elaborate synaptic strata enriched with synaptic proteins (*Figure 8C*). Dense arrangements of processes belonging to anti-5HT-positive afferent neurons (*Figure 8C*) correspond to intrinsic neuron layers (*Figure 8B*) and arrangements of anti-TH-positive terminals (*Figure 8D*). Thin processes of intrinsic neurons belonging to the two calyces extend centrally as two separate columnar lobes Clm1 and Clm2.

The column provided by Ca1 (Clm1) demonstrates an orthogonal network – characteristic of insect and stomatopod mushroom bodies – of intrinsic cell processes intersected by afferent and efferent neurons and by modulatory elements (*Figure 9B–E,H,I*). The columns contain dense arrangements of synaptic specializations (*Figure 9F,G*) and although these do not clearly distinguish successive synaptic domains, the arrangement of anti-TH-positive processes reveals this organization, one of the diagnostic characteristics of the mushroom body ground pattern (*Figure 9I*). The smaller of the two calyces provides intrinsic fibers to its columnar extension (Clm2), which terminates as a system of tubercles corresponding to the organization of the mushroom bodies in basal insects (Zygentoma, Ephemeroptera). These also show dense arrangements of synaptic sites (*Figure 9J*, inset).

## Background

The family Crangonidae is here represented by the sand shrimp *Crangon franciscorum* and the horned shrimp *Paracrangon echinata*, belonging to the superfamily Crangonoidea, which originated at the end of the Cretaceous (*Davis et al., 2018*). Both species provide interesting comparisons with the alpheids with respect to their habitat and mode of life. *Paracrangon* looks superficially like an almost colorless version of *Lebbeus*. However, unlike *Lebbeus*, which is highly active and hunts on rocky substrates and reefs, *P. echinata* is an almost immobile nocturnal ambush predator confined to crevasses. Its relative inactivity may be attributable to having only four walking legs compared to *Lebbeus*'s six (*Jensen, 2011*).

## Observations

The lateral protocerebrum of *Lebbeus* (*Figure 10A*) or *Spirontocaris lamellicornis* (*Figure 10B*) is radically different from that of *Paracrangon echinata* (*Figure 10C*), which possesses a single but substantial calyx that has an intense affinity to anti-DC0. Its cytoarchitecture appears to be generally homogeneous apart from a subtle indication of a peripheral layer immediately beneath its rostral covering of globuli cells (*Figure 10C*). The calyx gives rise to a diminutive system of anti-DC0-positive parallel fibers that provide a narrow column terminating in a small tubercular cluster (*Figure 10D*).

The last caridean considered here is *Crangon franciscorum,* which inhabits subtidal sandy habitats where it actively preys on smaller shrimp species, amphipods and other microfauna (*Siegfried, 1982*). The calyx of *C. franciscorum* shows low affinity to anti-DC0 but gives rise to a prominent system of anti-DC0-positive columns that terminate in swollen tubercles (*Figure 11A*). The entire system of the calyx, lobes and adjacent neuropils, including the visual system's lobula, is densely packed with anti-GAD-positive processes (*Figure 11B*). The calyx possesses stratified arborizations, resolved by anti-5HT and anti-TH immunostaining, some of which suggest subdivisions laterally across the calyx (*Figure 11C,D*). In this species, several of the lateral protocerebral neuropils caudal to (beneath) the

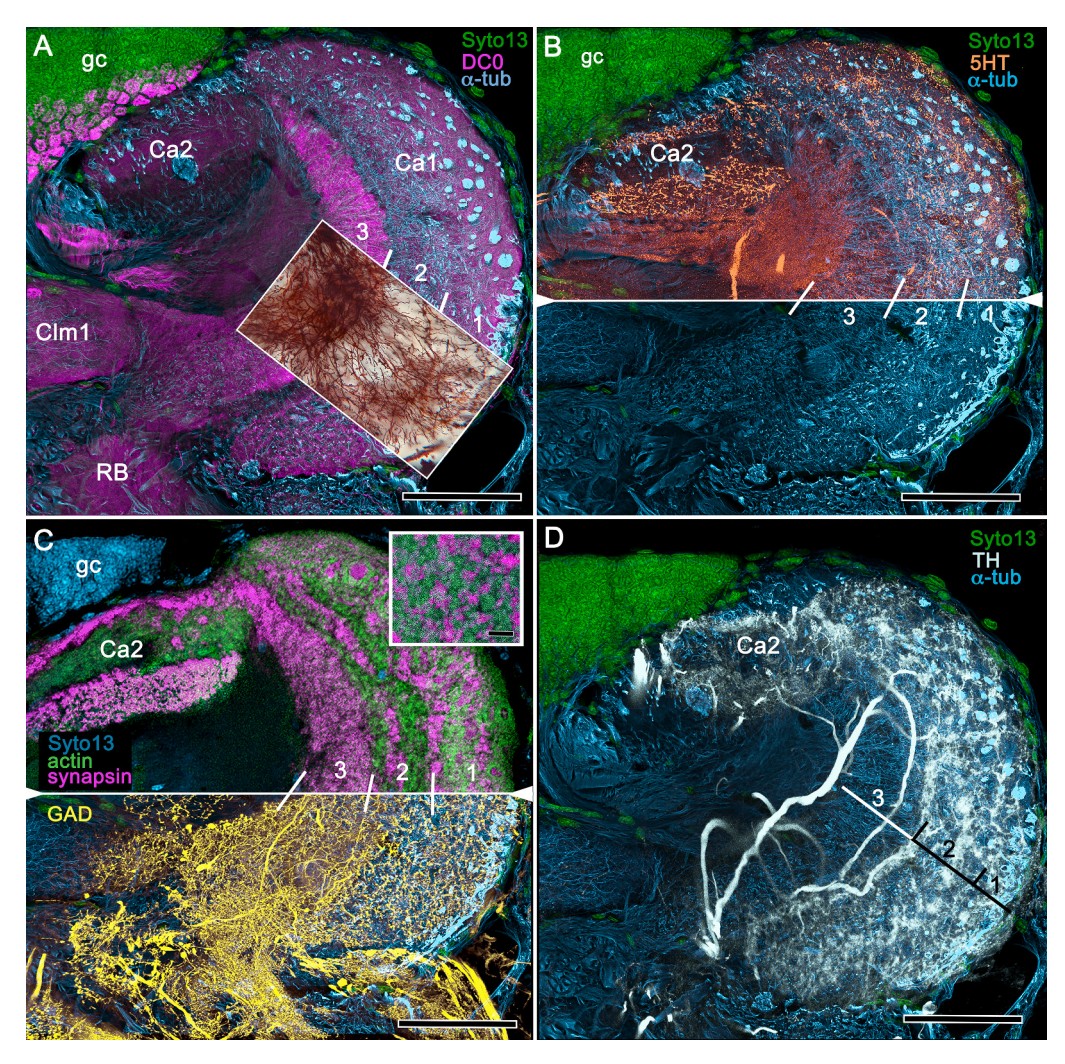

**Figure 8.** Neuroanatomy of mushroom body calyces in the thorid shrimp, *Lebbeus groenlandicus*. (A-D) Confocal laser scans of immunohistochemically labelled sections (anti-α-tubulin is cyan in all panels except C). (A) Anti-DC0 (magenta) immunoreactivity in both calycal regions (Ca1, calyx 1; Ca2, calyx 2), as well as in the columnar extension of intrinsic cells (Clm1) and the reniform body (RB). The inset demonstrates a palisade of Golgi-impregnated intrinsic cells, the dendritic organization of which reflects the presence of three distinct layers in Ca1 (1–3 against inset). (B) Anti-5HT immunolabelling (orange) showing afferent neuron terminals from various regions in the lateral protocerebrum ending in both calyces, and the three layers in Ca2 (upper half of panel). Calycal cytoarchitecture is distinguishable by anti-α-tubulin labelling alone (lower half of panel). (C) Upper half. Double labelling with anti-synapsin (magenta) and F-actin (green) further resolves layering (1-3) of synaptic sites in both Ca1 and Ca2. Globuli cells (gc) are shown in cyan. Inset. High-resolution scans resolve synaptic microglomeruli. Lower half. Anti-GAD-immunopositive fibers (yellow) extending from the lateral protocerebrum to provide dense innervation into all calycal levels. (D) Large-diameter anti-TH-positive terminals (white) spread their tributaries throughout calycal layers 1 and 2. Scale bars in A-D, 100 μm.

calyx appear to comprise many neuropil islets, each with as strong an affinity to anti-DC0 as has the calyx. The disposition of these islets suggests a system possibly corresponding to the reniform body (*Figure 11A*).

## Clade Reptantia

Reptantia are estimated to have originated in the mid-late Devonian (*Porter et al., 2005*; *Wolfe et al., 2019*). They are a natural group comprising genera that all possess a novel satellite

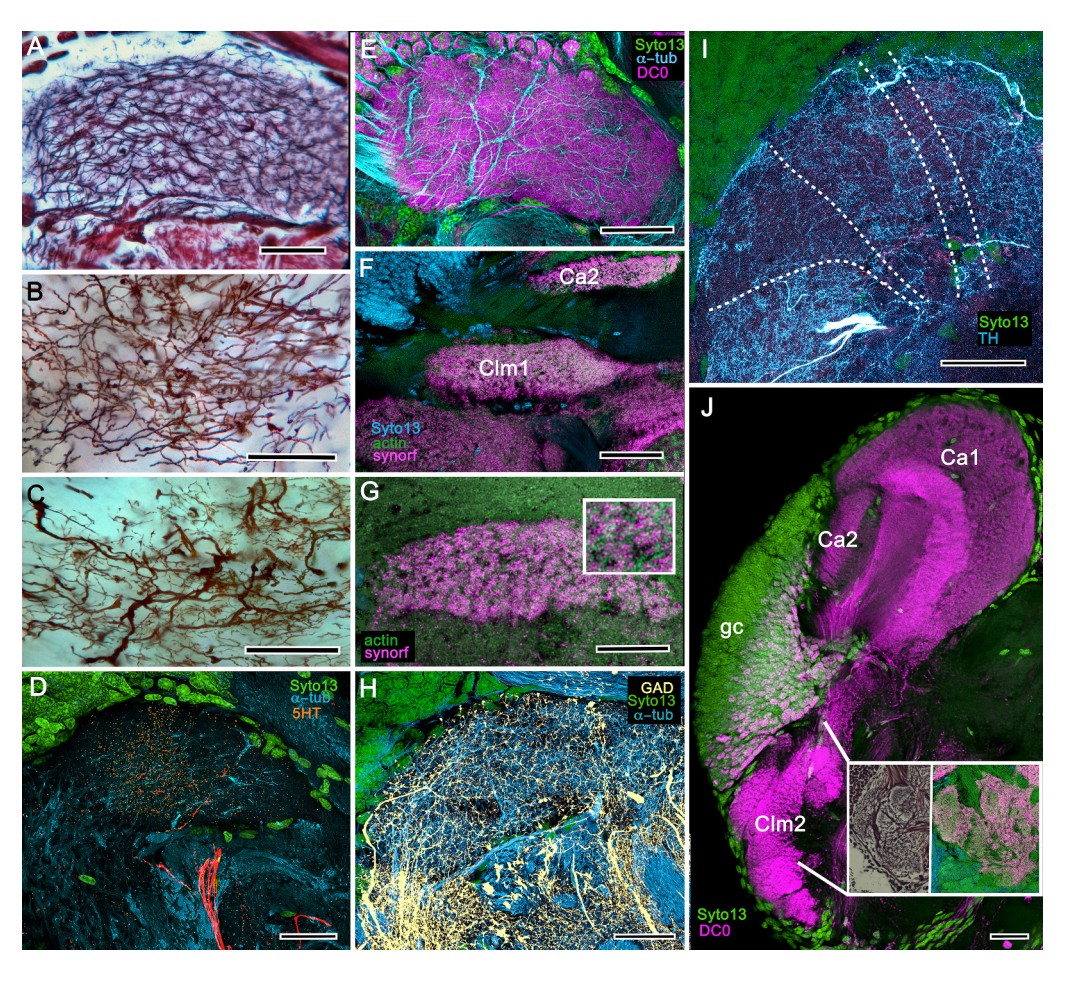

**Figure 9.** Columnar projections of *Lebbeus* mushroom body intrinsic neurons. (**A**) Bodian-stained axon-like extensions of intrinsic neurons are not parallel but interweave throughout the length of the first column (Clm1). (**B**). Golgi impregnations resolve these decorated by bouton- and spine-like specializations denoting, respectively, afferent pre- and postsynaptic specializations. (**C**) Golgi impregnations resolve afferent and efferent (MBON) processes intersecting serial domains within Clm1. (**D**) Sparse anti-5HT-immunoreactive neurons (orange) send terminals into Clm1 from regions in the lateral protocerebrum. (**E**) Anti-DC0 (magenta) immunoreactivity reveals a high affinity for this antibody in Clm1 where it outlines afferent and α-tubulin-positive efferent fibers (cyan). (**F**) Anti-synapsin (magenta) expression in Clm1 and Ca2., (**G**) Double labelling with anti-synapsin (SYNORF1) and F-actin (green) reveals microglomeruli in Clm1 (inset). (**H**) Anti-GAD-positive processes (yellow) project along the full length of Clm1. (**I**) Discrete serial domains along Clm1 are defined by processes of putative MBONs immunoreactive to anti-TH (cyan). (**J**) Anti-DC0 immunoreactivity (magenta) defines Clm2 showing its characteristic terminal tubercles (compare with *Figure 10A*). The two insets resolve tubercles in Bodian silver-stained material (left) and anti-synapsin/F-actin (magenta/green) labelling of their synaptic zones (right). Scale bars in A-J, 50 μm.

neuropil in the deutocerebrum, called the accessory lobe adjacent to the olfactory lobe (*Sandeman et al., 1993*).

## Transformed mushroom body in recent astacids

### Background

Fossil and geologically calibrated molecular phylogenies place the origin of the infraorder Astacidea, the lineage providing what are generally known as 'lobsters' and 'crayfish' at 250–400 mya (*Wolfe et al., 2019*). Here we identify derivations of the ancestral mushroom body in the lateral protocerebra of the North American freshwater species *Orconectes immunis* (calico crayfish) and *Procambarus clarkii* (Louisiana crawfish).

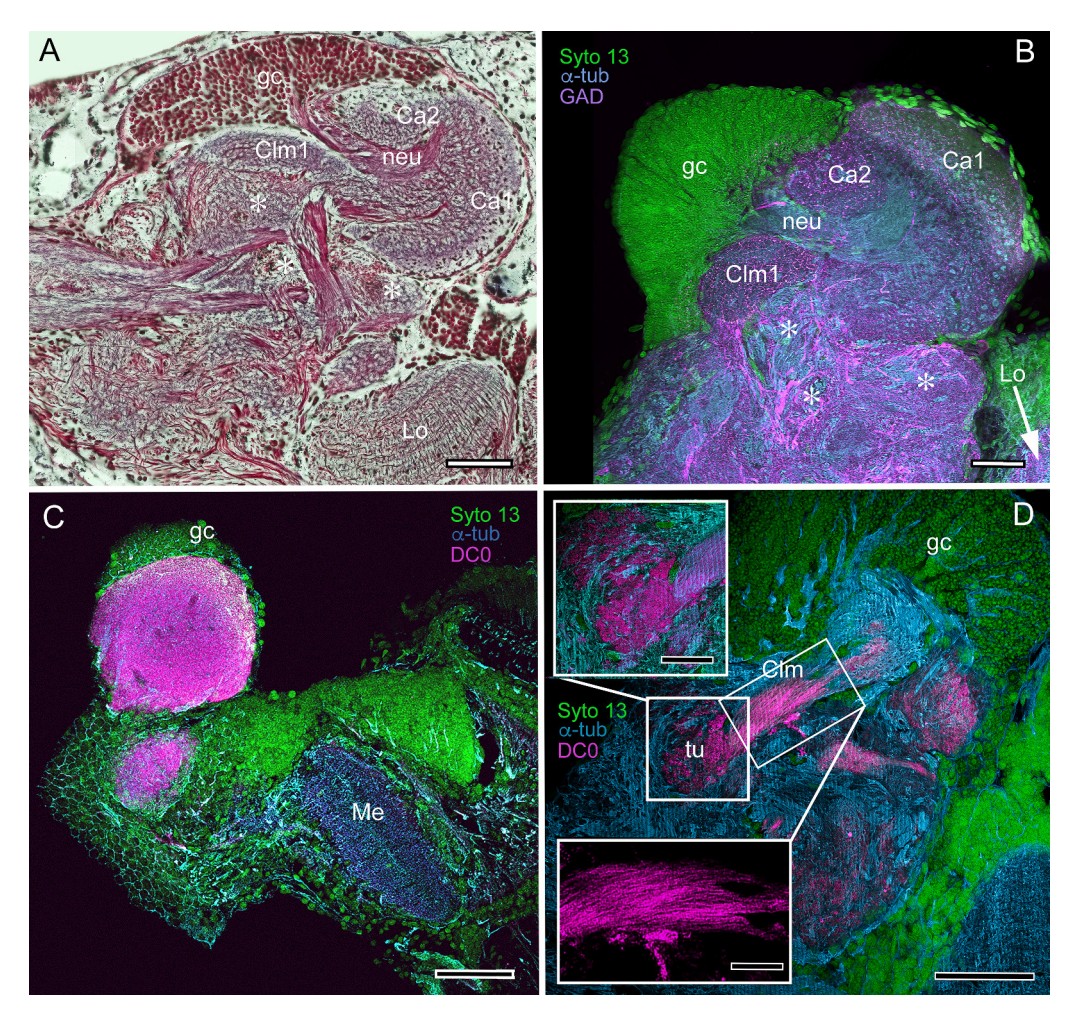

**Figure 10.** Mushroom body neuroarchitecture: *Spirontocaris lamellicornis* and *Paracrangon echinata*. (A) Bodian-stained section of *Lebbeus groenlandicus* demonstrates that the organization of the mushroom body calyces and columns of this thorid shrimp is almost identical to that of species sharing the same ecological niche, exemplified by the thorid *S. lamellicornis* and the crangonidid *Paracrangon echinata*. (B) Anti-α-tubulin (cyan) and anti-GAD immunoreactivity (magenta) in *S. lamellicornis* resolve column 1 (Clm1), the two calyces Ca1, Ca2, and their neurites (neu) leading from globuli cells (gc; green, Syto13). More caudal regions of the corresponding lateral protocerebrum are indicated by asterisks in panels A and B. (C) Anti-DC0 immunostaining (magenta) of *P. echinata* reveals high levels of the antigen in a substantial rounded calyx with a narrowly differentiated outer stratum covered by a dense layer of globuli cells (gc, green). (D) A second specimen showing evidence of a diminutive column (Clm), resolved by anti-DC0 (magenta), comprising parallel fibers (inset, lower left) ending in a small tubercular (tu) domain (inset, upper left). gc globuli cells (green, Syto13); Me medulla; Lo lobula. Scale bars in A-D, 100 μm; insets in D, 50 μm.

## Observations

The description by *Sullivan and Beltz (2001)* of the hemiellipsoid body of *Procambarus clarkii* identified its overall organization as two layered volumes, which we have named the cupola and torus (*Strausfeld and Sayre, 2020*). Neither volume appears outwardly distinct, the matrix of each comprising many microglomeruli (*Figure 12A*). However, closer inspection shows that these are larger and more densely packed in the torus than the cupola (*Figure 12A*, inset). Golgi impregnations, described in *Strausfeld and Sayre (2020)*, resolve further differences between these two neuropils: each possesses a constrained organization of intrinsic neurons, which have their dendrites and short axon-like processes limited to either the cupola or torus. Both neuropils contain elaborate systems

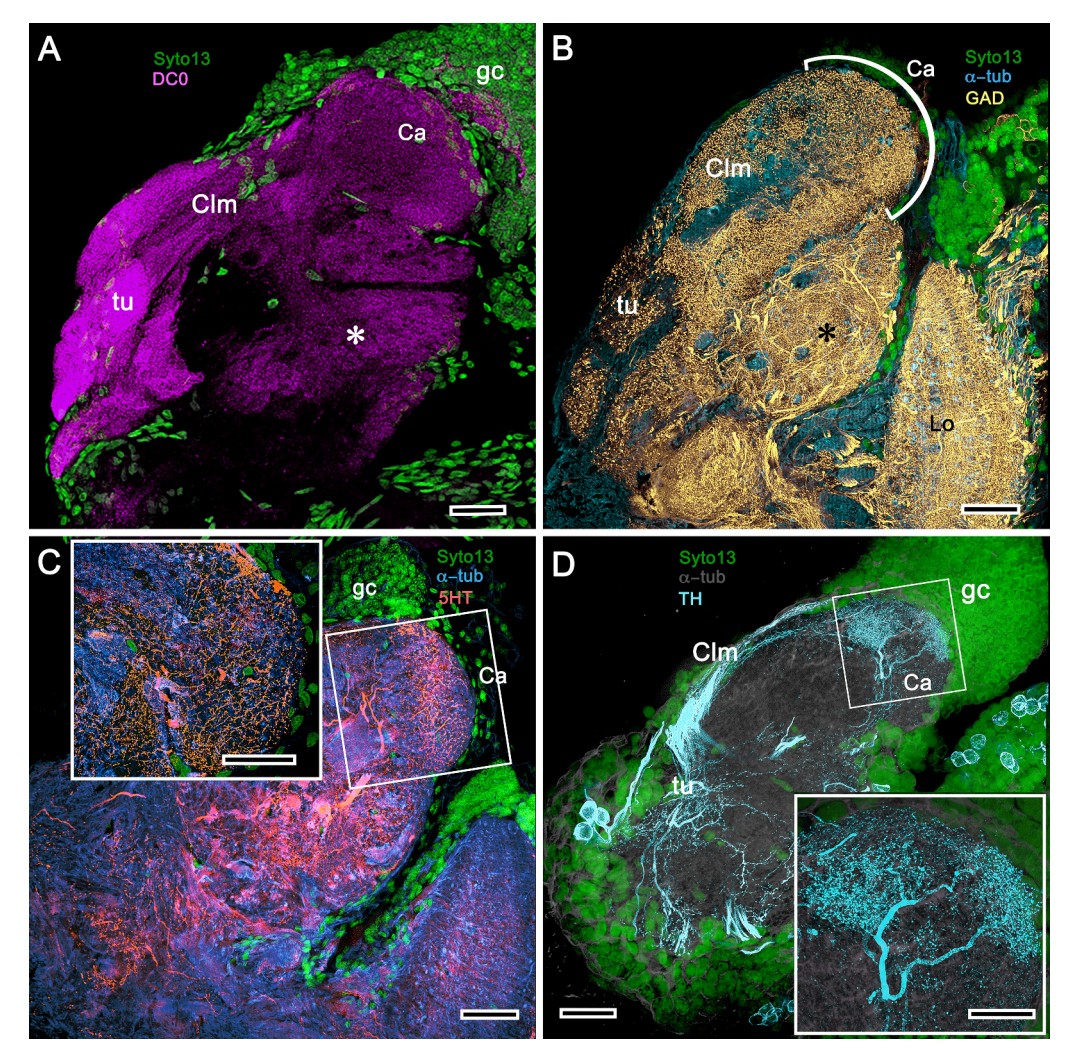

**Figure 11.** Mushroom body of *Crangon franciscorum*. (**A**) Immunohistochemically labelled sections reveal moderate anti-DC0 expression (magenta) in the mushroom body calyx (Ca) of *C. franciscorum* but a system of intensely labelled columnar lobes (Clm) ending as tubercular swellings (tu). As in carideans and reptantians, this species reveals lower levels of anti-DC0 immunoreactivity (asterisk) in the more distal parts of the lateral protocerebrum, beneath the calyx. (**B**) Anti-GAD immunoreactivity (yellow) is abundant in the calyx and the tubercles, but is sparser in the columnar lobes. Anti-GAD immunoreactivity also extends throughout the lateral protocerebrum and lobula (Lo). (**C**) Anti-5HT labelling (orange) reveals putative serotoninergic processes in the calyx (inset: details shown in single scan) and regions within the lateral protocerebrum. (**D**) Putative dopaminergic neurons, labelled with anti-TH (cyan), branch in two distinct territories of the calyx (Ca), enlarged in the inset lower right (gray, α-tubulin). gc globuli cells (green, Syto13). Scale bars in A-D, 50 μm, both insets, 20 μm.

of local interneurons and glomerular ensembles, suggestive of local circuits, orthogonal arrangements appearing more pronounced in the torus (see also *Figure 12—figure supplement 1*). Anti-DC0 immunoreactivity shows both cupola and torus to be usually, but not invariably, highly immunoreactive (compare *Figure 12C* with *Figure 12D,E*), as are discrete regions beneath them, here referred to as the subcalycal neuropil area VII *Figure 12C* (nomenclature from *Blaustein et al., 1988*) and, laterally, the reniform body (RB, *Figure 12C*). Depending on the angle of sectioning, the outer level is dome-like – hence the nomenclature cupola – whereas the inner level appears to have a toroidal architecture. These volumes are matched by the distribution of anti-GAD immunoreactivity (*Figure 12E*). Organization of the cupola and torus as two independent units is also matched by the distribution of profuse anti-TH-immunoreactive efferent neurons and more sparsely distributed anti-5HT-immunoreactive processes (*Figure 12F–H*). Recordings from these neurons show them to be

multimodal output neurons, comparable to MBONs of mushroom body columns (*Mellon, 2000*; *Mellon and Alones, 1997*; *McKinzie et al., 2003*).

## Mushroom body hypotrophy of the ghost shrimp *Neotrypaea californiensis* (*Figure 12—figure supplement 2*)

### Background
*Neotrypaea californiensis* is a species of ghost shrimp belonging to the 150–300 mya (Triassic) infra-order Axiidea (*Wolfe et al., 2019*). This species inhabits shallow, intertidal waters covering muddy substrates in which it constructs deep burrows (often shared by individual *Betaeus harrimani*). Fossils of the subfamily Callianassinae, to which *Neotrypaea* belongs, first appear in the Eocene (56 mya; *Hyžný and Klompmaker, 2016*).

### Observations
Labelling brains of the ghost shrimp *Neotrypaea californiensis* with anti-DC0 reveals paired mushroom body-like centers, which, because the species lacks eyestalks, are contained in foreshortened lateral protocerebra entirely within the head. As described in an earlier study (*Wolff et al., 2012*), where these centers were referred to as hemiellipsoid bodies, their neuropil is divided into two stacked volumes. The lower is intensely labelled by anti-DC0, whereas the upper volume, which comprises many hundreds of microglomerular synaptic sites, is not (*Figure 12—figure supplement 2*). Thus, this center has clear mushroom body attributes: a microglomerular calyx surmounting a condensed and foreshortened column, the neuropil of which is associated with a cluster of anti-DC0-immunoreactive perikarya in an adjacent globuli cell cluster.

## The chimeric mushroom body of Paguroidea and its evolved derivative

### Background
Originating in the Devonian, Reptantia gave rise to two branches, one providing the infraorders Achelata, Polychelida and Astacidea, the other providing the infraorders Axiidea and Gebiidea, and infraorders Anomura and Brachyura (together Meiura). Fossil-calibrated molecular phylogenies (*Wolfe et al., 2019*) demonstrate the superfamily Paguroidea within the infraorder Anomura diverging in the late Triassic to provide six families of which two, Paguroidae and Coenobitidae, are marine and land hermit crabs, respectively. Coenobitidae is the younger, diverging approximately 35 mya. Here we compare anti-DC0-immunoreactive centers in the lateral protocerebrum of the marine hairy hermit crab *Pagurus hirsutiusculus* with those of the land hermit crab *Coenobita clypeatus* (*McLaughlin et al., 2010*).

### Observations
In *Pagurus*, affinity to anti-DC0 (*Figure 13A*) reveals two immediately adjacent calycal domains in the lateral protocerebrum that together are continuous with an anti-DC0-positive columnar lobe. The neural architecture of the calyces (*Figure 13B–D,F–H*) is remarkably similar to the organization of afferents and intrinsic neurons in successive layers of the multistratified anti-DC0-immunoreactive hemiellipsoid body of the land hermit crab *Coenobita clypeatus* (*Figure 13I–O*). As demonstrated by Golgi impregnations (*Wolff et al., 2012*; *Strausfeld and Sayre, 2020*), both *Pagurus* and *Coenobita* have corresponding networks provided by a variety of morphological types of intrinsic neurons. These form precisely defined orthogonal networks (*Figure 13B,C*) of dendritic processes and terminals. In *Pagurus*, one population of intrinsic neurons provides axon-like extensions into the lobes (*Strausfeld and Sayre, 2020*). In *Coenobita*, all of the axon-like processes of intrinsic neurons are constrained to within the layered arrangements of its hemiellipsoid body (*Wolff et al., 2012*). In *Pagurus*, the dendrites of anti-5HT- and anti-TH-immunoreactive MBONs intersect intrinsic neuron terminals in the tubercular swellings at the end of the columnar lobes. The layered arrangements of the dendrites of these immunoreactive neurons reflect the repeat organization of precisely aligned intrinsic neuron dendrites across each calyx's width (*Figure 13C,E*).

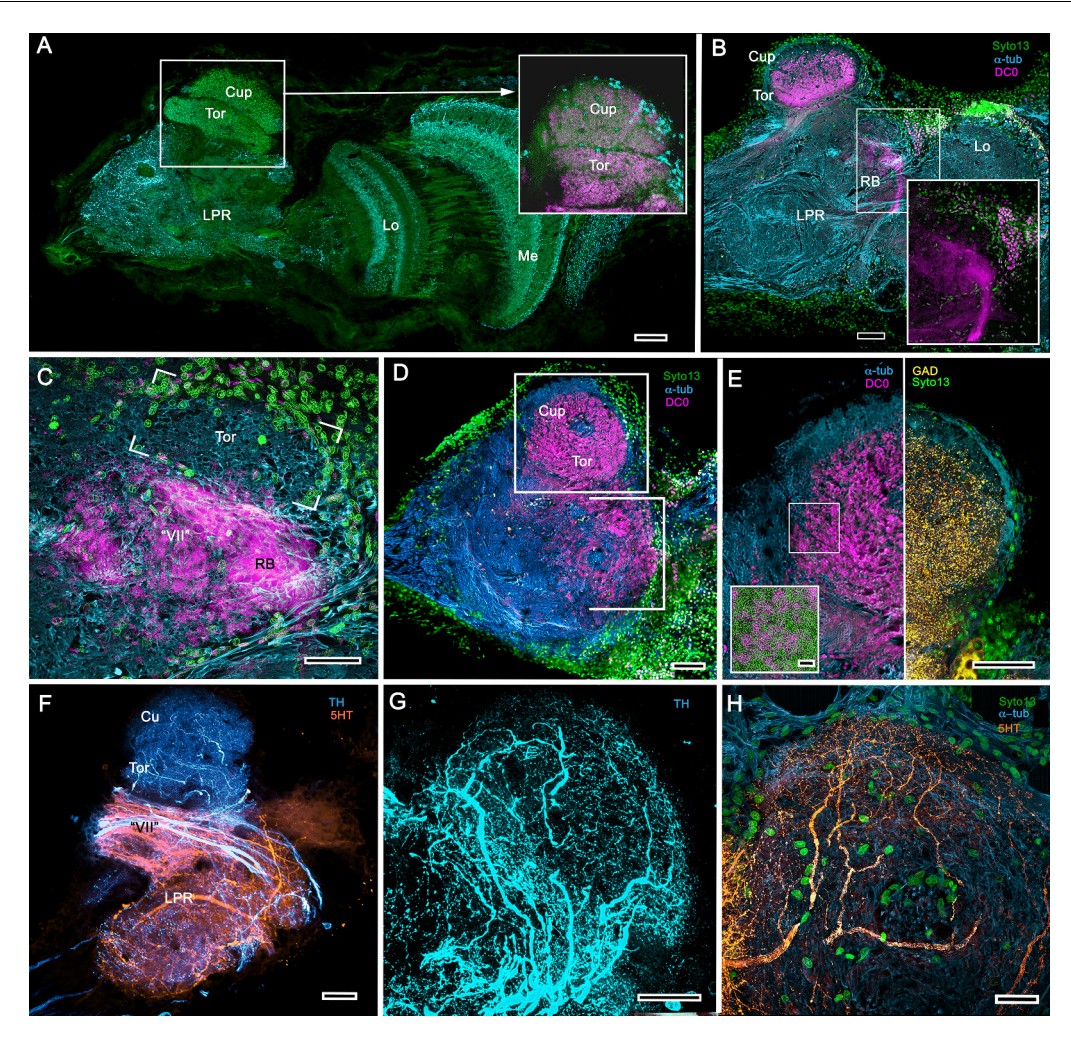

**Figure 12.** Transformed mushroom bodies in Astacidea. (**A**) An overview of the crayfish lateral protocerebrum showing its distinctive hemiellipsoid body composed of two components, the rostral cupola (Cup) overlying the torus (Tor). All panels show the right lateral protocerebrum, where rostral is upwards and distal is to the right (F-actin, green; anti-allatostatin, cyan). Inset to A. anti-synapsin and F-actin staining (magenta/green) shows the torus equipped with denser synaptic clusters than the cupola above it. However, it is the neuropil beneath the torus, receiving a massive input from the olfactory lobes, that shows the greatest synaptic density. (**B**) Anti-DC0 immunolabelling (magenta) here shows the torus as more densely labelled. Neuropils ascribed to the reniform body (RB) are further distal, at the border between the lateral protocerebrum (LPR) and the optic lobe's lobula (Lo). The inset shows a cluster of anti-DC0-positive cell bodies at the rostral surface associated with the reniform body like those identified in Brachyura (see *Figure 15E*, (**F**). (**C**) A section just glancing the torus, here showing almost no anti-DC0 immunoreactivity in the torus, but substantial anti-DC0 labelling in sub-calycal neuropils corresponding to neuropil (VII) described by *Blaustein et al. (1988)* and the probable location of the reniform body (RB). (**D**) Intense anti-DC0 labelling of both levels of the hemiellipsoid body and neuropils beneath and distal to this center (bracketed). (**E**) Alignment of the anti-DC0-positive hemiellipsoid body in D to the left with that of another specimen immunolabelled with anti-GAD (yellow) to the right. Discrete small glomerulus-like aggregates resolved by anti-DC0 (corresponding area enlarged in inset) contrast with the uniform distribution of GAD-immunoreactive profiles. The inset in E (left) demonstrates structures corresponding to the anti-DC0-immunolabelled aggregates comprise anti-synapsin/F-actin labelled (magenta-green) synaptic microglomeruli comparable to those identified in the stomatopod mushroom body calyces (*Wolff et al., 2017*). (**F–H**) Antibodies against TH (cyan) and 5HT (orange) demonstrate distributions of these efferent neurons, which correspond to multimodal parasol cells (efferent neurons equivalent to MBONs) described by *Mellon (2003)* and *McKinzie et al. (2003)*. Scale bars in A, B, D, F, 100 µm; E, G, H, 50 µm; inset to E, 2 µm.

The online version of this article includes the following figure supplement(s) for figure 12:

*Figure 12 continued on next page*

*Figure 12 continued*

**Figure supplement 1.** Neuronal arrangements in the hemiellipsoid derivative.
**Figure supplement 2.** Mushroom body hypotrophy of the ghost shrimp *Neotrypaea californiensis.*

## The hemiellipsoidal mushroom body of *Coenobita clypeatus*
### Background
The anomuran family Coenobitidae, to which the Caribbean land hermit crab belongs, is a recent group appearing approximately 35–40 mya (*Bracken-Grissom et al., 2013*; *Wolfe et al., 2019*). *Coenobita clypeatus* and its cousin the coconut crab *Birgus latro* have attracted considerable attention regarding olfactory adaptations associated with terrestrialization (*Stensmyr et al., 2005*). However, the juveniles develop in a marine habitat and, as expected, their antennular olfactory receptor neurons are typical of marine species in possessing ionotropic receptors, the axons of which terminate in large deutocerebral olfactory lobes (*Harzsch and Hansson, 2008*; *Groh et al., 2013*).

### Observations
As shown in both *B. latro* and *C. clypeatus,* massive relays from the olfactory lobes project to the lateral protocerebra to terminate in greatly inflated centers referred to as hemiellipsoid bodies (*Harzsch and Hansson, 2008*). These neuropils are defined by numerous anti-DC0-positive stratifications (*Figure 14A,B*). Golgi impregnations and electron microscopy demonstrated that these tiered arrangements consist of orthogonal networks of intrinsic neurons, postsynaptic to afferent terminals and presynaptic to dendrites, resulting in an organization that corresponds to circuitry recognized in the insect mushroom body's columnar lobe (*Brown and Wolff, 2012*; *Wolff et al., 2012*). Volumes of neuropil situated laterally beneath the dome are also anti-DC0-positive, as is a globular center corresponding to the reniform body (*Figure 14A,B*). The correspondence of this center's internal organization with that of a mushroom body's column is further demonstrated by the arrangements of dendritic trees belonging to output neurons, which are equivalent to MBONs. *Figure 14C* shows that anti-TH- and anti-5HT-immunoreactive dendrites extend across strata, each of which is also defined by synaptic configurations revealed by F-actin and anti-synapsin (*Figure 14D*) and, in *Figure 14F*, by anti-α-tubulin and an antibody against PKA-RII that regulates DC0 (see Methods). Antibodies against GAD reveal broadly distributed processes that are not aligned within rows but extend across them (*Figure 14C,E*).

## Extreme divergence of the mushroom body in Brachyura
### Background
Brachyura (true crabs) is a species-rich infraorder, now comprising about 6800 species (*Ng et al., 2008*), which diverged 240 mya (*Wolfe et al., 2019*). Here we consider two species, both of which spend intertidal periods out of water, the shore crab *Hemigrapsus nudus* (Varunidae) and fiddler crab *Uca* (*Minuca*) *minax* (Ocypodidae). Fossil-calibrated molecular phylogenies indicate a mid-to-late Cretaceous origin for the Varunidae and Ocypodidae (*Tsang et al., 2014*; *Wolfe et al., 2019*).

The opportunistic detritivore *H. nudus* lives in a flat visual ecology interrupted by occasional pebbles and rocks. These shore crabs often come into immediate contact with each other, where they may initiate contact-induced actions, such as defensive aggression involving jousting and pushing, but there is no obvious ritualized courtship as there is in fiddler crabs. Individuals employ crevices as transitory residences, responding to predatory threats by opportunistically retreating and hiding in them (*Jacoby, 1981*). It is likely that they can deploy allocentric memory, judging from the habit of near- or fully-grown individuals to retreat to a specific location when threatened. These actions contrast with those of fiddler crabs, such as *Uca minax* – deposit feeders living on mudflats – that are renowned for their ritualized courtship displays and rapid path integration to established home sites to escape predation (*Crane, 1975*), and visual behaviors that relate to compound eyes adapted for vision in a flat world (*Zeil and Hemmi, 2006*).

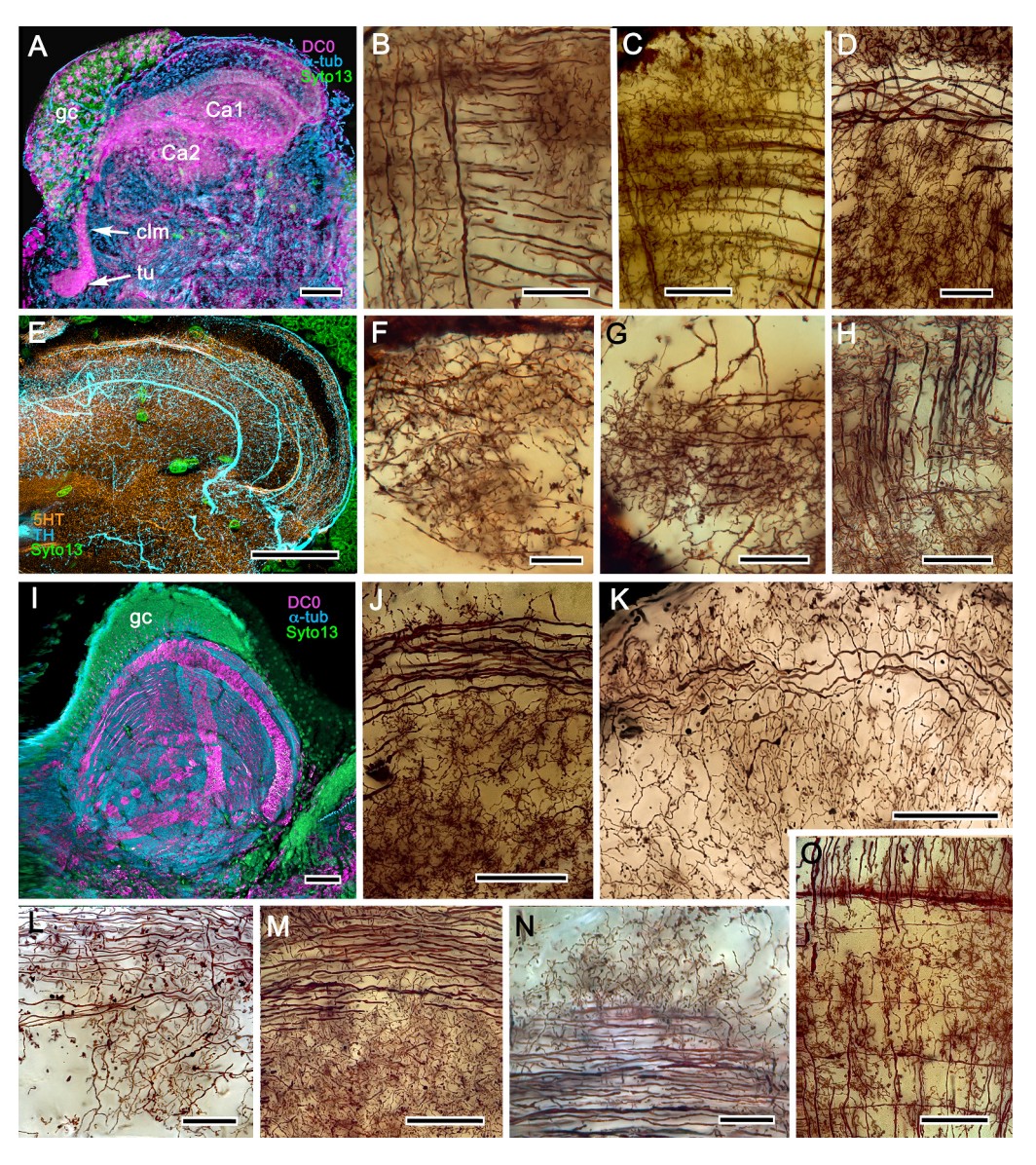

**Figure 13.** The reptantian mushroom body with and without its columnar lobe. Golgi impregnations demonstrate corresponding orthogonal arrangements in the two calyces of *Pagurus* (Ca1, Ca2 in panel **A**), which extend as a columnar lobe (clm) ending in tubercular swellings (tu), and in the multistratified hemiellipsoid body of *Coenobita* (**I**), which lacks a lobe. (**B-D**), (**F-H**) Golgi impregnations of *Pagurus* calyces show characteristic orthogonal arrangement of intrinsic neuron dendrites (**B,F,G**) originating from parallel arrangements of their initial processes (**C**). These neurons are invested by beaded fibers from the olfactory globular tract (OGT; **F**) and provide axon-like elongations that extend to the lobe (**D**; see also, *Strausfeld and Sayre, 2020*). Anti-5HT (orange) and anti-TH (cyan) immunostaining resolves corresponding efferent dendritic trees aligned with orthogonal networks (**E**). These are shown for the multistratified *Coenobita* hemiellipsoid body (panel **I**). (**J–O**) Golgi impregnations of the *Coenobita* hemiellipsoid body demonstrate orthogonal and rectilinear organization of its intrinsic neurons. Scale bars in A, 50 μm; B-D, 25 μm; E, 50μ; F, 10 μm; G, 25 μm; H,J,K, 50 μm; I, 100 μm; L-O, 25 μm.

## Observations

Treatment of eyestalks of six *H. nudus* with anti-DC0 resulted in four of twelve eyestalks showing barely detectable immunoreactivity with one specimen exhibiting some immunoreactivity in a tributary of the OGT. Eight consistently showed various levels of immunoreactivity in regions of the

rostral lateral protocerebrum. *Figure 15A–F* shows three of five specimens that all resolved strong immunoreactivity of large parts of the rostral neuropil associated with clusters of globuli cells lying over the lateral protocerebrum's rostral surface. One cluster of globuli cells also supplies a well-defined anti-DC0-immunoreactive neuropil positioned immediately proximal to the lobula. This matches the elaborations of the reniform body (*Figure 15E*), as resolved in Golgi impregnations and Bodian reduced-silver-stained specimens (*Thoen et al., 2019*). This center is known to participate in visual habituation and was understandably (but mistakenly) interpreted by *Maza et al. (2016)* as a mushroom body. As shown by *Thoen et al. (2019)*, this center is not a homologue of the mushroom body or the hemiellipsoid body but is present with either of those centers. Depending on the tilt of sectioning and the section's depth the intensely anti-DC0-immunoreactive pedestal of the reniform body can be resolved extending obliquely from the rostro-dorsal to the caudal margin of the lateral protocerebrum (*Figure 15A–F*). Sections of fortuitously oriented blocks can reveal the entire reniform body complex, here shown digitally enhanced against other anti-DC0 territories in *Figure 15E*.

*Figure 15* depicts territories of the lateral protocerebrum of *H. nudus* indicating that almost the entire rostral half of the neuropil is occupied by very large anti-DC0-immunoreactive domains characterized by cortex-like boundaries of fissures and lobes (*Figure 15B*). In *H. nudus*, the volume associated with the anti-DC0-labelled domain is relatively free of anti-GAD immunoreactivity, whereas antibodies raised against 5HT and TH suggest that anti-TH may be more abundant in the lateral protocerebrum's upper half than in the lower (*Figure 15G,H*). The fiddler crab, *Uca minax* (*Figure 15— figure supplement 1*), reveals similar anti-RII-immunoreactive territories that also appear to be highly folded. Less extensive than those in *H. nudus*, the two most immunoreactive territories are separated by a volume showing lower affinity to anti-DC0. Notably, vibratome sections treated with SYNORF1 (anti-synapsin) and phalloidin (F-actin) to resolve synaptic microglomeruli, selectively reveal only specific subterritories of the volume occupied by the anti-DC0-positive center (inset: *Figure 15—figure supplement 1* inset).

Scrutiny of anti-DC0 territories identifies anti-synapsin as well as anti-TH- and anti-5HT-immunoreactive arrangements suggestive of mushroom body-like configurations (*Figure 15I,J*), the locations of which correspond to the boxed area of *Figure 15D*.

## Discussion

### Mushroom body transformation: Hemiellipsoid bodies

Our results demonstrate substantial variation of the mushroom body ground pattern in pancrustaceans where it has undergone major transformations in Reptantia (*Figure 16*), a monophyletic group for which fossil-calibrated molecular phylogenies estimate a time of origin at 400–385 mya (*Wolfe et al., 2019*; *Porter et al., 2005*). Reptantians are distinguished by their apomorphic accessory lobes (*Hanström, 1925*; *Hanström, 1931*), spherical or crescent-shaped neuropil comprising diminutive synaptic islets that receive inputs from the adjacent olfactory lobe (*Sandeman and Luff, 1973*; *Wachowiak et al., 1996*). In reptantian infraorders Achelata, Astacidea, and Axiidea the accessory lobe supplies inputs to the 'hemiellipsoid bodies' (*Hanström, 1925*; *Hanström, 1931*), which are distinct from the 'standard' mushroom body morphology. In Achelata and Astacidea these consist of two well-defined neuropils lacking columnar lobes, arranged side by side (as in spiny lobsters) or one over the other (as in crayfish; *Blaustein et al., 1988*; *Sullivan and Beltz, 2004*; *Sullivan and Beltz, 2005*). In Axiidea they are single highly condensed neuropils.

The great majority of descriptions have focused on hemiellipsoid body neuropils (e.g., *Derby and Blaustein, 1988*; *Wachowiak and Ache, 1994*; *Mellon and Alones, 1997*; *Mellon, 2003*; *McKinzie et al., 2003*; *Schmidt and Mellon, 2010*) some promoting them as apomorphic equivalents (e.g., *Sandeman et al., 2014*; *Machon et al., 2019*). However, two lines of neuroanatomical evidence support hemiellipsoid bodies as mushroom body homologues. First, their organization reflects an evolved transformation of the ancestral ground pattern, in which the columnar lobe neuropil has been subsumed into the calyx, which in Reptantia comprises elaborate networks of short-axon and anaxonal intrinsic neurons, and parallel fibers. These are associated with anti-DC0-positive synaptic microglomeruli, comparable to calycal microglomeruli reported for the stomatopod and hexapod calyces (*Wolff et al., 2017*; *Groh and Rössler, 2011*).

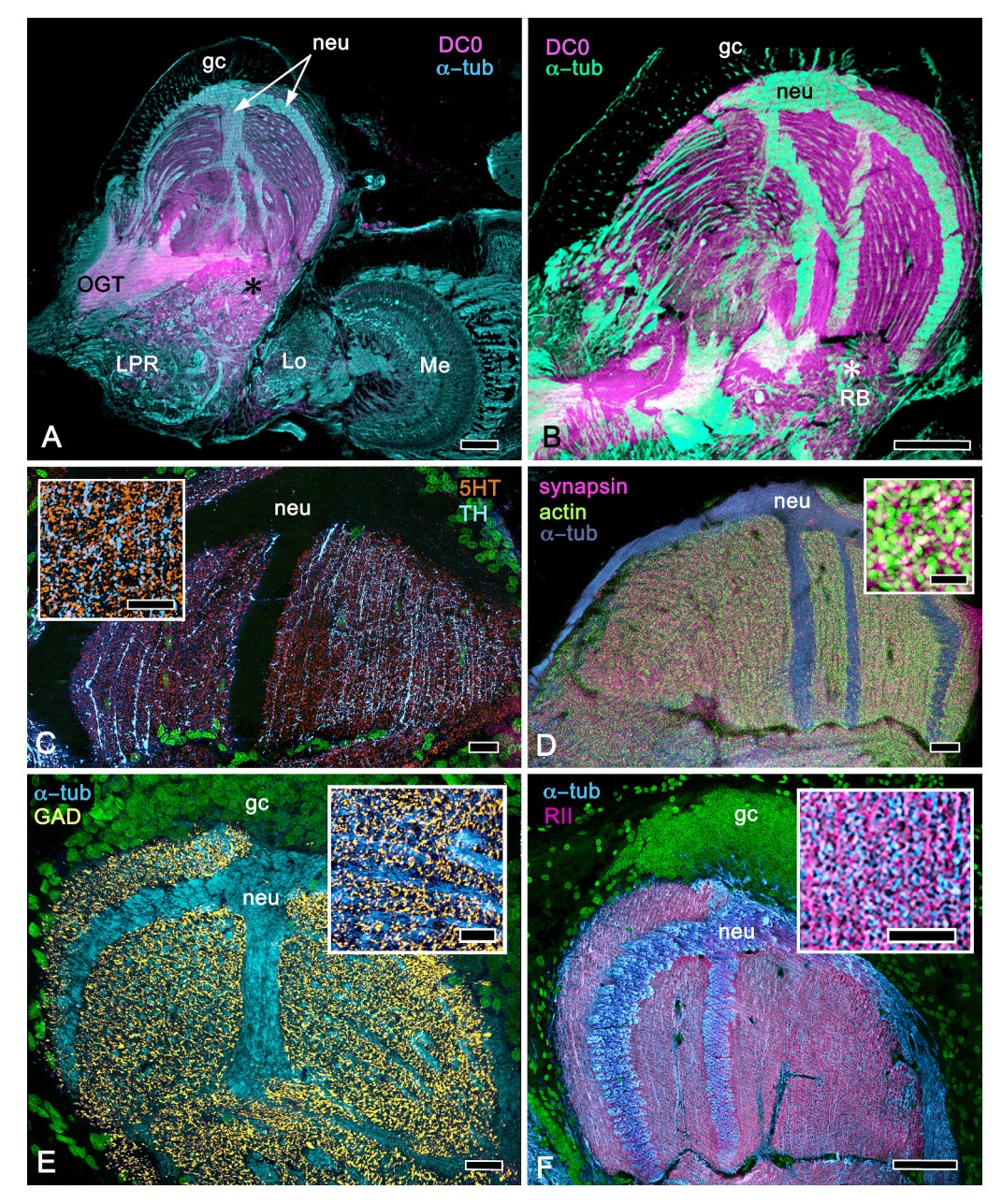

**Figure 14.** Calycal hypertrophy in the *Coenobita clypeatus* mushroom body. (**A,B**) DC0 and other immunological markers reveal a characteristic system of nested strata comprising orthogonal fibers that originate from the intrinsic neurons (neu; cyan, anti-α-tubulin). Anti-DC0 (magenta) defines these strata as well as the entrance of the olfactory globular tract (OGT) into the calyx and several regions of the lateral protocerebrum beneath the calyx, including the reniform body (RB, asterisk in A,B). Other regions of the lateral protocerebrum (LPR,) and optic lobe (Lo, lobula; Me, medulla) show little or no anti-DC0 affinity. (**C**) Anti-5HT- (orange) and anti-TH-immunoreactive (cyan) fibers in the calyx extend within each of the anti-DC0-positive strata but are notably absent from the neurite bundles of intrinsic neurons. The distribution of anti-TH is not uniform (inset), but patchy indicating discrete domains of TH-immunopositive processes within the stratified intrinsic neuron networks. (**D**) Anti-synapsin (magenta) and F-actin (green) demonstrate regions of dense synaptic connectivity indicated by microglomerular configurations (inset). In this panel, intrinsic neuron bundles (neu) are labelled with anti-α-tubulin (cyan). (**E**) Distributed processes labelled with antisera against GAD (yellow) extend across strata, revealing neuropil occupied by afferents to the calyces (inset). (**F**) RII, an antiserum developed in *Drosophila melanogaster* against the regulatory subunit of PKA, confirms an expression pattern (magenta) corresponding to that of DC0 (**A,B**) and is shown comingling with anti-α-tubulin labelled neuropil in and between stratifications (inset). Scale bars in A,B, 100 μm; C-E, 20 μm; F, 50 μm; inset C, 5 μm; inset D, 2 μm; insets E-F, 5 μm.

Second, outputs (parasol cells) relaying multisensory associations from the hemiellipsoid bodies correspond to mushroom body output neurons (MBONs) (*McKinzie et al., 2003*; *Mellon et al., 1992*). Like MBONs, parasol cells, which target other regions of the lateral protocerebrum, are resolved by antisera against TH and 5HT (*Figure 12F–H*).

These astacid arrangements contrast with centers, also referred to as hemiellipsoid bodies (*Krieger et al., 2010*), in the reptantian lineage Anomura. In the marine hermit crab, *Pagurus hirsutiusculus*, two adjoining calyces are composed of a hybrid arrangement of intrinsic neurons contributing to stratified arrangements of orthogonal networks corresponding to arrangements observed in the columnar mushroom body lobes of insects (*Strausfeld and Sayre, 2020*). Studies of these circuits in the more recent Coenobitidae have demonstrated that their synaptic organization corresponds to that of an insect mushroom body's column (*Wolff et al., 2012*; *Brown and Wolff, 2012*). In land hermit crabs (*Birgus latro* and *Coenobita clypeatus*) although the hemiellipsoid bodies are greatly inflated, matching the huge olfactory lobes that define these species' deutocerebra (*Krieger et al., 2010*), they nevertheless possess the same morphological attributes as *Pagurus*, except that their stratified arrangements are reiterated many times over to provide stacked synaptic strata. In both marine and land hermit crabs, large efferent neurons, identified by their immunoreactivity to anti-5HT and anti-TH, insinuate dendrites that extend across and between these stratifications to provide yet another example of neurons that correspond to MBONs leading to protocerebral centers. Hemiellipsoid bodies in the squat lobster *Munida quadrispina*, belonging to the superfamily Galatheoidea, which shares a common ancestor with Paguroidea, have a comparable arrangement of strata (*Hanström, 1931*; *Strausfeld and Sayre, 2020*), indicating this as a defining character of Anomura.

In true crabs (Brachyura) identification of a possible mushroom body homologue is problematic. The few recent studies on the brains of crabs include descriptions of their optic lobes (*Sztarker et al., 2005*) and reniform body (*Thoen et al., 2019*), in addition to the brains of the shore crab *Carcinus maenason*, and species of fully terrestrial crabs (*Krieger et al., 2012b*; *Krieger et al., 2015*). However, no clearly defined center, comparable to that of an astacid or anomuran 'hemiellipsoid body,' has been convincingly identified.

Anti-DC0 immunohistology of the shore crab *Hemigrapsus nudus* reveals a prominent rostral immunoreactive domain, denoted by fissures and lobes, occupying almost the entire rostral half of the lateral protocerebrum (*Figure 15*) immediately adjacent to the reniform body. This corresponds to the adjacent arrangement of the mushroom body and reniform body seen in Stomatopoda (*Thoen et al., 2019*). In the fiddler crab *Uca minax* (*Figure 15—figure supplement 1*), two immunoreactive territories similarly occupy a rostral location in the lateral protocerebrum.

These observations do not align with studies based on synapsin immunohistology approximatiing where hemiellipsoid bodies with astacid-like substructures might be (*Krieger et al., 2015*). And their designation as astacid-type hemiellipsoid bodies implies derivation from a dome-like ancestral morphology (*Krieger et al., 2012b*; *Krieger et al., 2015*). There is as yet no evidence for this, and the anti-DC0-immunoreactive volumes in Brachyura suggest a folded neuropil reminiscent of anti-DC0-immunoreactive centers of 'whip spiders' belonging to the arachnid order Amblypygi (*Wolff and Strausfeld, 2015*).

Arachnids offer a paradigm for an evolutionary scenario where basal and later evolving taxa demonstrate ancestral morphologies, whereas intermediate lineages may show highly derived morphologies. In Arachnida (*Sharma et al., 2014*; *Giribet, 2018*), the more basal Solifugae and Scorpiones possess 'standard' anti-DC0-immunoreactive mushroom bodies with a columnar lobe comprising parallel intrinsic neurons intersected by afferent and efferent arborizations (*Wolff and Strausfeld, 2015*). The most recent arachnids, 'whip scorpions' (Thelyphonida), also possess 'standard' mushroom bodies, whereas the intermediate Amblypygi have greatly enlarged and highly folded anti-DC0-immunoreactive centers. Arachnida have evolved modifications departing from, as well as retaining, even in the most recent lineages, the ancestral mushroom body ground pattern.

A similar evolutionary scenario appears to have occurred across Caridea and Reptantia where the ground pattern may reappear in younger lineages. In Crangonidae, *Crangon franciscorum* has a mushroom body-like calyx and columnar lobes, whereas its sister species *Paracrangon echinata* has a bulbous hemiellipsoid body-like center and a diminutive lobe, indiscernible unless labelled with anti-DC0 (*Figures 10C,D* and *11*). *Crangon* and *Paracrangon*, which occupy different habitats (see below), are estimated to have diverged in the early Eocene, about 56 mya (*Davis et al., 2018*). In

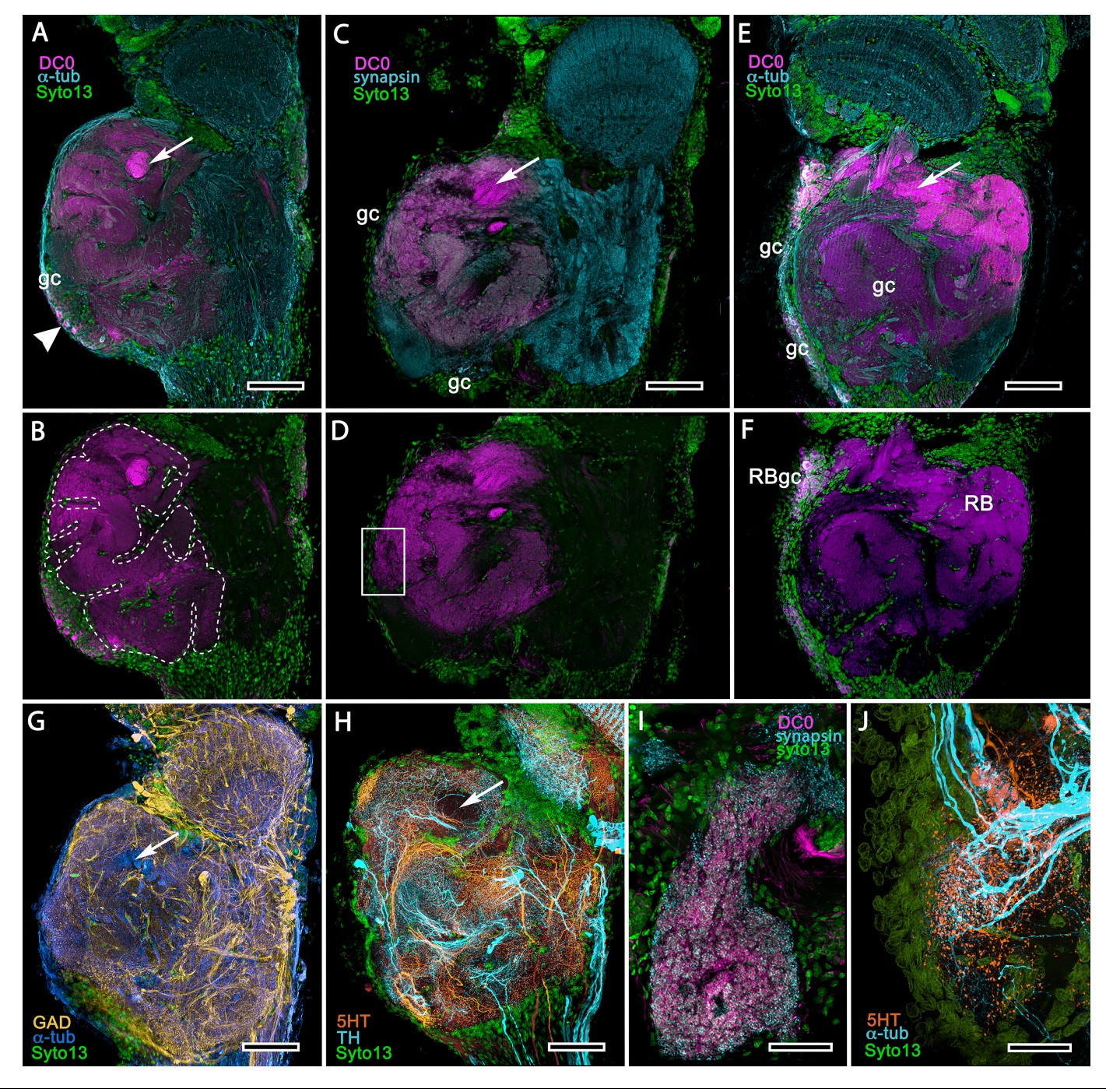

**Figure 15.** Mushroom body homologues of the shore crab *Hemigrapsus nudus*. (**A-H**) Depictions of sections of the right lateral protocerebrum: rostral is to the left; distal (towards the optic lobes) is the top of each panel. (**A–F**) Anti-DC0-immunoreactive territories (magenta) are interpreted as modified mushroom bodies, lacking columnar lobes and enormously expanded in the rostral lateral protocerebrum. Distally, each of these centers is penetrated by the pedestal (bright magenta in A-F, arrow in panels A,C,E,G,H) belonging to the large reniform body typifying varunid crabs. The entire territory occupied by the reniform body (RB) is shown in E against the dimmed magenta surround. Anti-DC0 labelling is shown at normal intensities in panels A-D, F. The maximum extent of the mushroom body neuropil is indicated by the dotted outline in B. Cell bodies of intrinsic neurons (gc, green: Syto13) occur scattered over the rostral and dorsal surface of the lateral protocerebrum (**A–F**), some also showing elevated anti-DC0 immunoreactivity (e.g., arrowhead in A). Globuli cells providing the reniform body (RBgc) also show anti-DC0 immunoreactivity, as indicated in panel F. (**G**) Distribution of anti-GAD-immunoreactive (yellow) processes mainly in caudal volumes of the lateral protocerebrum and the lobula. (**H**) Distribution of anti-5HT- (orange) and anti-TH-immunoreactive (cyan) processes. (**I,J**) Labelling with anti-DC0 (magenta) and the synaptic marker SYNORF1 (cyan; I) and with anti-5HT

*Figure 15 continued on next page*

*Figure 15 continued*

(orange) and anti-TH (J) reveal discrete regions (one boxed in D) in anti-DC0-immunoreactive territories that are suggestive of mushroom body-like circuitry. Scale bars in A-H, 100 μm; I, J, 25 μm.

The online version of this article includes the following figure supplement(s) for figure 15:

**Figure supplement 1.** Anti-DC0-immunoreactive neuropils (magenta) of the fiddler crab *Uca minax* (rostral to the left, distal upwards).

reptantian Anomura, there has been both evolved loss (Munididae, Coenobitidae) as well as retention (Paguridae) of the mushroom body's columnar lobe (*Strausfeld and Sayre, 2020*) during divergence times spanning 200 million years (*Wolfe et al., 2019*).

## Mushroom body transformation: Reduction and loss

Second-order olfactory integration centers may not be required in species that live in conditions where associative memories or place memories are of negligible relevance. The minute anti-DC0-immunoreactive centers of *Penaeus*, shown in *Figure 3A*, suggest evolved reduction in a species that spends part of its adult life in the featureless ecology of the off-shore water column (*Bauer, 2019*). In certain isopods, possibly olfactory specialists, the mushroom body is reduced to a mere vestige, as in *Figure 3B*. In terrestrial isopods, absence of a hemiellipsoid body has been suggested to relate to diminution of the olfactory pathway (*Harzsch et al., 2011*). Complete loss is also reported for species constrained to unusual ecologies, such as fresh water caverns (*Ramm and Scholtz, 2017*; *Stegner et al., 2015*). The marine isopod *Saduria entomon* also seems to have a greatly reduced hemiellipsoid body (*Kenning and Harzsch, 2013*). But as cautioned by *Ramm and Scholtz (2017)*, that the olfactory globular tract (OGT) terminates in a much reduced lateral protocerebrum does not imply the presence of a hemiellipsoid body.

Nevertheless, miniaturizing of the mushroom body is exemplified in Leptostraca, sister to all Eumalacostraca (*Wolfe et al., 2016*). Until now, evidence for such a center has been tenuous (*Kenning et al., 2013*) and its absence would be problematic: if mushroom bodies and their modifications are a defining feature of Pancrustacea, then mushroom bodies would be expected to occur in this basal malacostracan group.

Application of anti-RII to the brain of the leptostracan *Nebalia pugettensis* identifies an immunoreactive neuropil at the location suggested by *Kenning et al. (2013)*. It comprises a relatively large shallow cap over the rostro-dorsal lateral protocerebrum (*Figure 17A,B*). The cap provides two very short columnar lobes, in which anti-α-tubulin and anti-RII immunoreactivity suggest parallel arrangements of axon-like processes typical of the mushroom body ground pattern (*Figure 17D,E*). Small perikarya, suggestive of globuli cells, are contiguous with a larger population also supplying optic lobe neuropils (*Figure 17A,B*). The smallness of the mushroom body in *Nebalia* suggests a possible evolved miniaturization, comparable to that described from hexapod Collembola (*Kollmann et al., 2011*). The mid-Silurian *Cascolus ravitis* is the oldest known fossil stem representative of this group (approximately 430 mya: *Siveter et al., 2017*), and leptostracans have since undergone morphological and probably behavioral simplification (*Dahl, 1985*). Extant species are mostly epibenthic scavengers or suspension feeders, many limited to burrowing habits in simple ecologies (*Martin et al., 1996*; *McCormack et al., 2016*).

Nonetheless, life in the water column though devoid of visual landmarks is rich in other sensory cues: tactile, vibrational, thermal and chemosensory. In certain predatory Copepoda, small intertidal members of Multicrustacea, glomerular antennular lobes are connected by heterolaterally projecting OGTs to prominent hemiellipsoid bodies located in their lateral protocerebra (*Andrew et al., 2012*).

Likewise, the allotriocarid Remipedia, which are also blind predatory crustaceans, have prominent anti-DC0-immunoreactive centers in both lateral protocerebra supplied heterolaterally by the OGT (*Stemme et al., 2016*). These centers give rise to a small columnar extension (see Figure 3 in *Stemme et al., 2016*), also visible in thin sections as a short stubby column extending from the larger neuropil (see Figure 6A,B in *Fanenbruck and Harzsch, 2005*). This is reminiscent of the arrangement in *Nebalia* (*Figure 17B,D,E*). Yet molecular phylogenies identify Remipedia as the closest crustacean relative of Hexapoda (*Oakley et al., 2013*; *Schwentner et al., 2017*) – a conundrum, because overall brain organization in Remipedia is typical of Malacostraca (*Fanenbruck et al., 2004*).

## Olfactory organization and mushroom body divergence

The primary sensory input to mandibulate mushroom bodies is olfactory, and the olfactory pathways of crustaceans and insects, as far as the lateral protocerebrum, have been promoted as homologous (*Schachtner et al., 2005*; *Harzsch and Krieger, 2018*). However, there are profound differences between the crustacean and insect olfactory systems. We propose here that these differences may underlie our finding that mushroom body morphology varies considerably across Malacostraca (*Figure 16*), whereas it is highly conserved across dicondylic insects (*Strausfeld et al., 2009*).

Differences between the crustacean and (dicondylic) insect olfactory system begin at the level of the uniramous appendages of the pancrustacean deutocerebrum: the crustacean paired antennules and the insect paired antennae. Although these are segmental homologues (*Boxshall, 1999*), the crucial difference between the two applies to their olfactory sensilla and olfactory receptor neurons (ORNs). Aesthetascs, the flexible odorant sensilla of crustaceans, are distributed in malacostracans on the antennule's lateral flagellum. Each sensillum may contain some hundreds of ORNs, the dendrites of which may further branch to provide extensive membrane surfaces (*Tierney et al., 1986*; *Grünert and Ache, 1988*; *Schmidt and Mellon, 2010*; *Hallberg and Skog, 2010*). In insects, the four principal morphologies of odorant sensilla can contain 1–6 poorly branched or unbranched ORNs (*Schneider and Steinbrecht, 1968*; *Altner and Prillinger, 1980*). An exception is the hymenopteran placode sensillum serving 30+ ORNs (*Getz and Akers, 1994*).

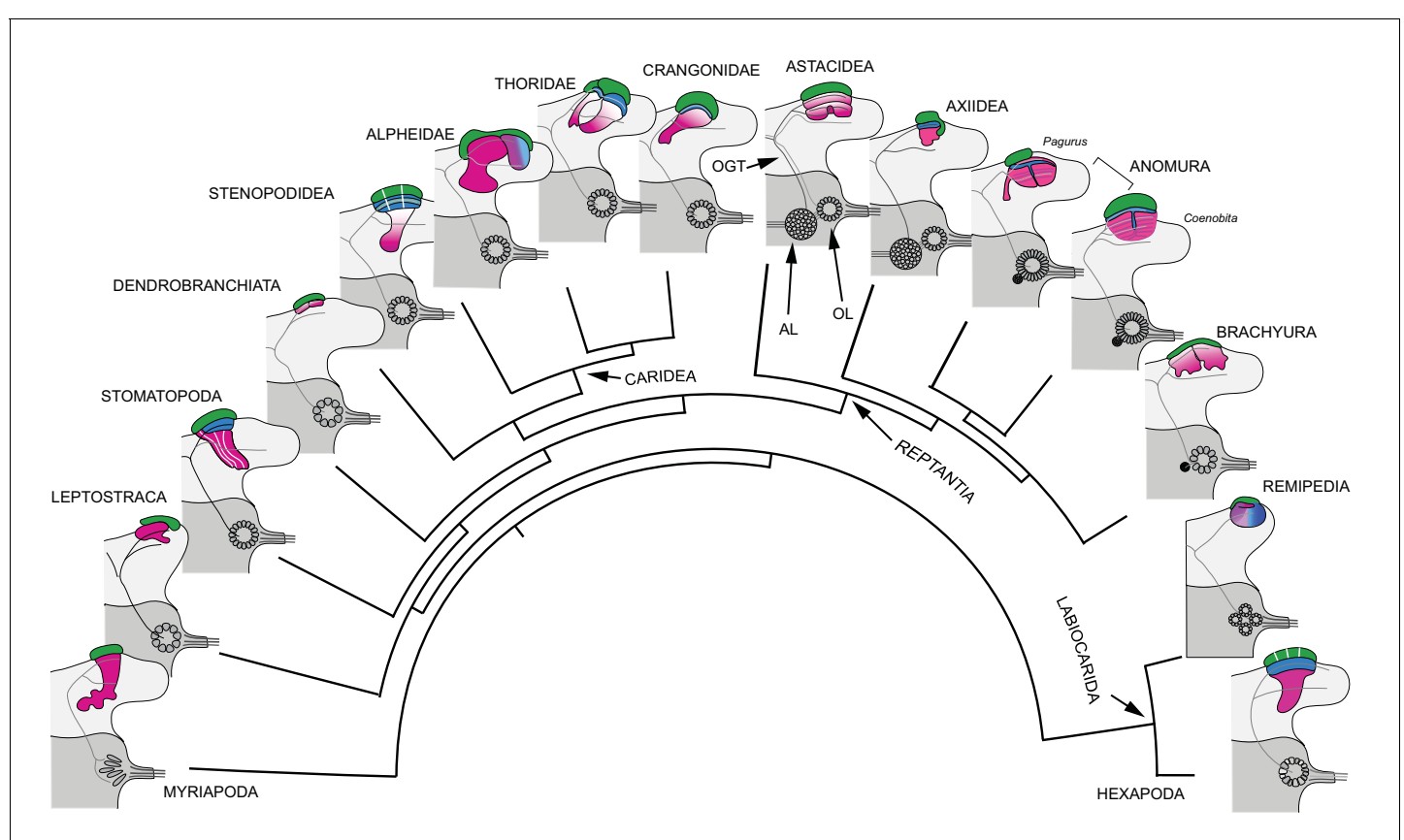

**Figure 16.** Retention and divergence of the mushroom body ground pattern in Mandibulata. Schematized shapes of mushroom bodies described in this study (for actual profiles see *Figure 1B*). Their evolved derivatives are mapped onto a pancrustacean molecular phylogeny (after: *Oakley et al., 2013*; *Wolfe et al., 2019*; *Schwentner et al., 2017*), here extended to include the mandibulate outgroup Myriapoda (represented by the chilopod mushroom body; *Wolff and Strausfeld, 2015*). Each schematic depicts the right lateral protocerebrum (light gray) without its optic lobes and the right deutocerebrum (dark gray). The deutocerebrum is shown with its olfactory lobe (OL); or, in Reptantia, the olfactory and adjoining accessory lobe (AL). The antennal globular tracts (OGT) are shown with their contralateral extension indicated in all examples except Myriapoda and Hexapoda, where the OGT is exclusively homolateral. Magenta indicates anti-DC0/RII identification of corresponding neuropils; green indicates globuli cell clusters; blue indicates distinct calycal organization. Evolved diminution is here shown for Dendrobranchiata. Despite miniaturization in Leptostraca, the RII-immunoreactive center reveals reduced columnar lobes. A comparable arrangement is resolved in Remipedia (see Fig. 3 in *Stemme et al., 2016*).

**Table 1.** Comparison of main morphological characters defining the crustacean and insect olfactory pathways.

| Character | Hexapoda (Dicondylia) | Eumalacostraca and Remipedia | Correspondence |
|---|---|---|---|
| Ligand-gated odorant receptors (ORs) | Likely ubiquitous | None reported | – |
| Ionotropic olfactory receptors (IRs) | Present | Likely ubiquitous | + |
| Olfactory neurons in each sensillum | 1-5 (in Dicondylia) | Numerous, >10 and can exceed 100s | – |
| Axons of olfactory receptor neurons | Target-defined glomeruli | Non-targeting: can innervate multiple glomeruli | – |
| Olfactory lobe subunits | Unique glomeruli, relating to genetic receptor identity | Discrete subunits, isomorphic, number unrelated to receptor identities | – |
| Accessory lobes | None | Reptantian lineages (possible remipede homologue) | – |
| OL projection neurons (PNs) | Most uniglomerular | All multiglomerular | – |
| Accessory lobe projection neurons (via OGT) | N/A | In Reptantia | – |
| Number of PN axons from olfactory lobe | 2–5 times the number of glomeruli | Some thousands, unrelated to glomerulus number | – |
| Projection neuron axon tract (OGT or AGT) | AGT stays ipsilateral | OGT bifurcates, ipsi- and contralateral. (Exception is OGT ipsilateral in Cephalocarida) | – |
| Protocerebral PN terminals | In MB calyx + other lateral protocerebraltargets | In MB calyx + other lateral protocerebral targets | + |

These receptor-level distinctions are amplified by differences of sensory transduction mechanisms. Crustacean ORNs are exclusively equipped with ionotropic channels (*Derby et al., 2016*; *Kozma et al., 2018*), whereas in dicondylic insects, in addition to ionotropic ORNs, the majority of ORNs employ ligand-gated channels, there being as many variants of these in a species as there are genes that encode them (*Robertson et al., 2018*).

Differences between insects and crustaceans are profound in the olfactory lobes, which in crustaceans comprise identical wedge- or lozenge-shaped synaptic glomeruli. ORN axons terminate in one or several of these (*Tuchina et al., 2015*), but there is no hard evidence that any specific type of ORN targets a specific subset of glomeruli. This contrasts with insects. As shown for *Drosophila*, axons from different genetically determined ionotropic ORNs target discrete volumes of the antennal lobes (*Silbering et al., 2011*; *Rytz et al., 2013*). In *Drosophila*, and insects generally, each glomerulus has a specific location (and size and shape) in its antennal lobe and is functionally unique in receiving converging terminals from, mainly, a single genetically identical set of ligand-gated ORNs (*Fishilevich and Vosshall, 2005*). There are as many glomeruli as there are genes encoding ligand-gated receptors (*Couto et al., 2005*). These arrangements comprise an odotypic map, a feature apparently not present in crustaceans.

Mushroom bodies are supplied by relay neurons (projection neurons: PNs) originating from the olfactory glomeruli. This feature is entirely different in crustaceans and insects. Hundreds – in some species thousands – of axons belonging to PNs extend rostrally from the crustacean olfactory lobes (or in certain reptantians also from their adjacent accessory lobes) to reach the lateral protocerebrum, many axons dividing into two tributaries extending to both sides. In crustaceans, PNs have wide-field dendrites extending to numerous glomeruli; no PN confines its dendrites to a single unit of the olfactory lobe (*Wachowiak and Ache, 1994*; *Schmidt, 2016*). This sharply contrasts with

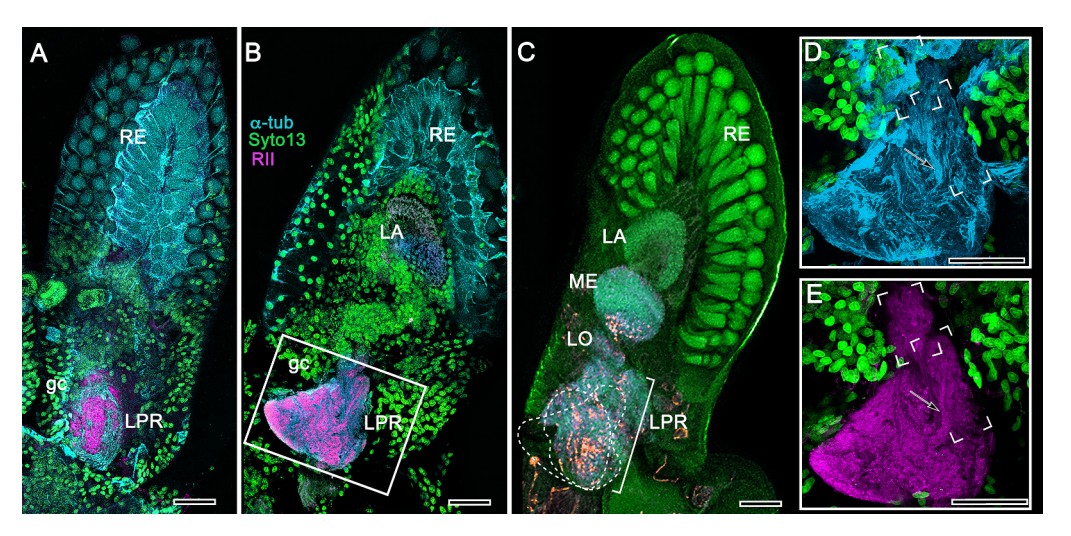

**Figure 17.** Mushroom body of *Nebalia pugettensis*. (A,B) Two successive sections reveal an ensemble of neuropils strongly immunoreactive to antibodies raised against RII (magenta, see Methods). An anti-RII-immunoreactive region of neuropil (A) expands into a planar calyx (B) and gives rise to two small but well-defined columnar lobes extending forwards. These regions (dotted lines) are superimposed in panel (C) beneath which are deeper levels of the lateral protocerebrum (LPR) comprising several discrete neuropils central to the nested optic lobe neuropils (LA, lamina; ME, medulla; LO, lobula). The boxed area in panel B is enlarged in (D, E). (D) Neuroarchitecture of the calyx resolved with anti-α-tubulin (cyan); the columnar lobes are bracketed, one showing bundled parallel fibers (arrowed in D,E). (E) Anti-RII-labeled (magenta) processes (arrowed) extending from the triangular calyx. RE, retina; gc, globuli cells. Orange in panel C is FMRFamide immunoreactivity. Scale bars indicate 50 μm.

insects where each glomerulus contains the dendritic trees of 3–5 PNs, and only a small population of additional PNs is associated with groups of glomeruli. Thus, relatively few PN axons extend rostrally from the insect antennal lobe and their axons exclusively target the ipsilateral protocerebrum to end in the mushroom body and, or, distally adjacent neuropils of the lateral horn (*Galizia and Rössler, 2010*). The only crustacean with comparable homolateral PN projections is the cephalocarid *Hutchinsoniella macracantha*, an allotriocarid basal to Branchiopoda, Hexapoda and Remipedia (*Stegner and Richter, 2011*; *Schwentner et al., 2017*).

As far as the lateral protocerebrum, each level of the olfactory pathway is distinctly crustacean or insect (*Table 1*). No crustacean possesses ligand-gated ORNs, as far as is known. In crustaceans, the arrangement amongst ORN terminals and widely branching PNs implies all-to-all connectivity in the olfactory lobe. There is, as yet, no evidence for functional differentiation of olfactory glomeruli, and it is currently assumed that reconstruction of chemical and temporal patterning of the odorant milieu is distinct from its encoding by the insect system (*Derby and Schmidt, 2018*). Molecular phylogenetics implies that the insect odorant receptor family (ORs; *Brand et al., 2018*) appeared after terrestrialization (*Missbach et al., 2014*) and that the 'labelled-line' organization of ORN terminals in specific insect olfactory ('antennal') glomeruli is an evolved innovation.

The contrast between the general uniformity of mushroom bodies in insects and their diversity of form in crustaceans may be considered in light of these distinctions between olfactory organization in insects and crustaceans. The suggestion here is that the 'labelled line' odotopic organization of the insect olfactory pathway from ORNs to antennal lobe glomeruli (*Grabe and Sachse, 2018*) may have constrained divergent evolution of the insect mushroom body, whereas the all-to-all connections amongst ORNs and PNs within the crustacean olfactory lobe (*Schmidt and Mellon, 2010*) may have permitted relaxed mushroom body evolution.

## Mushroom body divergence in Malacostraca and behavioral ecology

Along with differences in mushroom body morphology, ranging from columnar lobes to the hemiellipsoid body morphotype, pancrustacean species examined in this study also vary in their behavioral ecologies. The columnar lobes, and the calyces from which they protrude, are variously accentuated in certain genera, as exemplified by *Lebbeus groenlandicus* (Thoridae) and *Alpheus bellulus* (Alpheidae), whereas in others, circuits characteristic of the lobes are incorporated into the calyx to provide the dome-like structures Hanström referred to as hemiellipsoid bodies (*Hanström, 1925*; *Hanström, 1931*; *Hanström, 1932*).

Considering what is known about the habits of species investigated here, our observations suggest that highly mobile species occupying dynamic three-dimensional ecologies such as reefs have retained the mushroom body's columnar lobes. In insects, neurons associated with the columnar lobes provide continuous updating of spatial associations and their valences (*Cognigni et al., 2018*). Essentially, the aerial-terrestrial ecology exploited by dicondylic insects, all possessing mushroom bodies with columnar lobes, is comparable to elaborate three-dimensional ecologies of reefs and escarpments exploited by stomatopods and many caridean species likewise equipped with mushroom bodies having columnar lobes. The proposition is that crustaceans and insects possessing columnar mushroom bodies share the attribute of negotiating structurally elaborate three-dimensional habitats, often within defined territories. Ancestral mandibulates likely evolved in comparable ecologies. Micro-CT studies of their appendicular morphologies suggest that the oldest pancrustaceans from the lower Cambrian were locomotory adepts rather than simple animals that crawled on the seabed (*Zhai et al., 2019*). That mushroom bodies are crucial for spatial awareness is supported by studies demonstrating that the lobes increase in size during the acquisition of information about three-dimensional space (*Kühn-Bühlmann and Wehner, 2006*; *Montgomery et al., 2016*; *van Dijk et al., 2017*). Anomurans such as land hermit crabs, in which circuits characterizing mushroom body lobes have been subsumed into the calyces, have refined spatial memory for exploration and homing (*Krieger et al., 2012a*).

Large columnar mushroom bodies typify crustacean species that show place memory. Examples are cleaner shrimps such as *Stenopus hispidus*, motile reef dwellers such as *Lebbeus groenlandicus* or *Spirontocaris lamellicornis*, and active hunters such as Stomatopoda (*Wolff et al., 2017*; *Sayre and Strausfeld, 2019*). Life on the ocean floor, intertidal flats, or the bed of a stream or lake, such as that adopted by reptantian Achelata, Astacidea, and Axiidea, may require altogether different computational networks for coping with what is predominantly a planar world. Such distinctions are also suggested by recent Crangonidae, as discussed above.

## Morphological correspondence of the pancrustacean lateral protocerebrum

The entrenched opinion is that central to its optic lobe neuropils, the crustacean lateral protocerebrum comprises an integrative center called the 'medulla terminalis,' with which the hemiellipsoid body is invariably associated (see *Krieger et al., 2012b*; *Krieger et al., 2015*; *Krieger et al., 2019*). The original term 'masse médullaire terminale,' coined by the pioneering French neuroanatomist *Viallanes (1884)*, was to indicate numerous neuropils proximal to the optic lobes. However, *Fortuyn (1920)* used the term for a large (nonexistent) fourth optic neuropil and Hanström (*Hanström, 1925*; *Hanström, 1931*) used it to include all neuropils proximal to the optic lobes, usually without distinction except for the hemiellipsoid bodies.

The designation 'medulla terminalis' should be abandoned. First, it suggests a single synaptic neuropil rather than an elaborate organization of discrete centers. Second, it misleads by assuming the presence of a hemiellipsoid body-like center even when not resolvable (e.g. *Parhyale hawaiensis*: *Wittfoth et al., 2019*). Proposing equivalence of the 'medulla terminalis' with a specific center in the insect brain (e.g. the lateral horn: *Harzsch and Krieger, 2018*), could distract from deeper scrutiny. The lateral horn is a neuropil complex situated between the mushroom body calyces and the lobula where it is supplied by projection neurons from olfactory and optic glomeruli, (*Galizia and Rössler, 2010*; *Ito et al., 2014*; *Strausfeld, 1976*; *Strausfeld et al., 2007*). A possible, yet uncertain, crustacean homologue might be the reniform body, an integrative neuropil in a corresponding location that connects to the lobula and mushroom body in Stomatopoda and other Malacostraca (*Thoen et al., 2019*). That some eyestalk neuropils in Malacostraca likely correspond to those of the

insect lateral protocerebrum (*Strausfeld and Sayre, 2020*; *Ito et al., 2014*) is supported by *Blaustein et al. (1988)*, which, to date, is the only comprehensive study to name neuropil volumes that illustrate a level of elaboration comparable to the insect lateral protocerebrum; it is also the first study to appreciate Hanström's deployment of the term hemiellipsoid body.

### Mushroom bodies unify Mandibulata (Pancrustacea and Myriapoda)

Even as *Bellonci (1882)* identified and named the mantis shrimp's corpo emielissoidale, he stated that with its columnar extension – the corpo allungato – the two components were identical to the insect mushroom body. Likewise, Hanström's studies refer to these centers as corpora pedunculata, irrespective of whether they are pedunculate (*Hanström, 1925*; *Hanström, 1931*), stating that hemi-ellipsoid bodies are elaborated calyces. Yet, in the ensuing years the view that domed hemiellipsoid bodies of crustaceans are distinct from the lobed mushroom bodies of insects is specifically advo-cated as a basis for the ground pattern of the malacostracan brain (*Sandeman et al., 2014*; *Wittfoth et al., 2019*; *Machon et al., 2019*; *Krieger et al., 2019*).

The difficulty of relinquishing the hemiellipsoid body as a derived trait of crustaceans is exempli-fied by excluding the mushroom body's columnar lobes (compare the present *Figure 16* with *Machon et al. (2019)*; loc. cit. Figure 12), which is problematic for reaching consensus about pan-crustacean brain evolution. That mushroom bodies occur in Stomatopoda and caridean lineages necessitates reconsideration of the relationship between the mushroom body and structures referred to as hemiellipsoid bodies. One goal of the present paper has been to show how transformations from a columnar lobed morphotype to a calycal (hemiellipsoid) morphotype can be followed across eumalacostracan evolution (*Figures 1B* and *16*) while unifying this center in Eumalacostraca and Allotriocarida. Observations of Myriapoda (Diplopoda + Chilopoda), show this mandibulate group likewise defined by paired anti-DC0-immunoreactive mushroom bodies (*Wolff and Strausfeld, 2015*; *Wolff and Strausfeld, 2016*). Neuroanatomical traits identifying insect mushroom bodies fur-ther resolve these centers in Chelicerata and Onychophora and, across phyla, in spiralian Annelida and Platyhelminthes suggesting an ancient origin of this center (*Strausfeld et al., 2006*; *Wolff and Strausfeld, 2015*; *Wolff and Strausfeld, 2016*; *Strausfeld, 2019*).

It is indisputable that the diversity of crustacean mushroom bodies contrasts with the uniformity of mushroom bodies across Hexapoda other than Archaeognatha (*Strausfeld, 2012*) where evolved loss of the center is implied by Archaeognatha's phylogenetic position in Hexapoda (*Giribet, 2004*). No member of Hexapoda has been identified with an evolutionarily modified mushroom body com-parable to a hemiellipsoid body. The one fascinating exception is an experimentally induced point mutation that transforms the wild-type insect mushroom body to a hemiellipsoid body-like center. This is in the *Drosophila* brain mutant ks63 (mbd 'deranged'), the paired centers of which lack colum-nar lobes because their Kenyon cell axons are constrained to within the voluminous hybrid (calyx + lobe) neuropil, supplied by normal projection neurons from the olfactory (antennal) lobes. The mutant flies are able to discriminate odors and show odor-induced behavior, but were not shown to be capable of learning odors (*Heisenberg, 1980*; *Heisenberg et al., 1985*).

That the mushroom body ground pattern has persisted with far less variation across Hexapoda than it has across Crustacea is no reason to dispute that it unites Hexapoda and Crustacea. Evidence provided here demonstrates that in Crustacea mushroom bodies are – to paraphrase Richard Owen's famous statement (*Owen, 1843*) – the same center under every variety of form and function. The notion that hemiellipsoid bodies are something phyletically separate and distinct from mush-room bodies is based on a historical misapprehension that diverts attention from why such a plenti-tude of variations evolved and what drove their evolution.

## Materials and methods

**Key resources table**

| Reagent type (species) or resource | Designation | Source or reference | Identifiers | Additional information |
|---|---|---|---|---|
| Biological sample | *Stenopus hispidus* | Salty Bottom Reef Co. | SBRC-I-CBS | n = 25 |
| Biological sample | *Uca minax* | Salty Bottom Reef Co. | N/A | n = 12 |
| *Continued on next page* | | | | |

*Continued*

| Reagent type (species) or resource | Designation | Source or reference | Identifiers | Additional information |
|---|---|---|---|---|
| Biological sample | *Coenobita clypeatus* | Carolina Biological Supply Co. | 142415 | n = 13 |
| Biological sample | *Orconectes immunis* | Carolina Biological Supply Co. | N/A | n = 8 |
| Biological sample | *Procambarus clarkii* | Carolina Biological Supply Co. | 142512 | n = 10 |
| Biological sample | *Periplaneta americana* | Carolina Biological Supply Co. | 143642 | n = 20 |
| Biological sample | *Alpheus bellulus* | LiveAquaria | CN-90061 | n = 20 |
| Biological sample | *Stenopus hispidus* | LIveAquaria | CN-78439 | n = 3 |
| Biological sample | *Penaeus vannamei* | Gulf Specimen Marine Laboratories | N/A | n = 24 |
| Biological sample | *Neogonodactylus oerstedii* | Gulf Specimen Marine Laboratories | Ar-1600 | n = 25 |
| Biological sample | *Lebbeus groenlandicus* | Friday Harbor Laboratories | N/A | n = 37 |
| Biological sample | *Munida quadrispina* | Friday Harbor Laboratories | N/A | n = 2 |
| Biological sample | *Crangon franciscorum* | Friday Harbor Laboratories | N/A | n = 10 |
| Biological sample | *Spirontocaris lamellicornis* | Friday Harbor Laboratories | N/A | n = 3 |
| Biological sample | *Paracrangon echinata* | Friday Harbor Laboratories | N/A | n = 8 |
| Biological sample | *Neotrypaea californiensis* | Friday Harbor Laboratories | N/A | n = 11 |
| Biological sample | *Betaeus harrimani* | Friday Harbor Laboratories | N/A | n = 3 |
| Biological sample | *Hemigrapsus nudus* | Friday Harbor Laboratories | N/A | n = 40 |
| Biological sample | *Nebalia pugettensis* | Friday Harbor Laboratories | N/A | n = 4 |
| Biological sample | *Ligia pallasii* | Friday Harbor Laboratories | N/A | n = 3 |
| Biological sample | *Pagurus hirsutiusculus* | Friday Harbor Laboratories | N/A | n = 35 |
| Antibody | α-Tubulin (Mouse, monoclonal) | Developmental Studies Hybridoma Bank (DHSB) | CAT#: 12G10; RRID: AB_1157911 | 1:100 |
| Antibody | α-Tubulin (Rabbit, polyclonal) | Abcam | CAT#: ab15246; RRID: AB_301787 | 1:250 |
| Antibody | Synapsin (SYNORF1; Mouse, monoclonal) | Developmental Studies Hybridoma Bank (DHSB) | CAT#: 3C11; RRID: AB_528479 | 1:100 |
| Antibody | Serotonin (5HT; Rabbit, polyclonal) | ImmunoStar | CAT#: 20080; RRID: AB_572263 | 1:1000 |
| Antibody | Glutamic acid decarboxylase (GAD; Rabbit, polyclonal) | Sigma-Aldrich | CAT#: G5163; RRID: AB_477019 | 1:500 |
| Antibody | Tyrosine hydroxylase (TH; Mouse, monoclonal) | ImmunoStar | CAT#: 22941; RRID: AB_572268 | 1:250 |
| Antibody | DC0 (Rabbit, polyclonal) | Generous gift from Dr. Daniel Kalderon (*Skoulakis et al., 1993*) | RRID: AB_2314293 | 1:400 |

*Continued on next page*

*Continued*

| Reagent type (species) or resource | Designation | Source or reference | Identifiers | Additional information |
|---|---|---|---|---|
| Antibody | RII (Rabbit, polyclonal) | Generous gift from Dr. Daniel Kalderon (*Li et al., 1999*) | N/A | 1:400 |
| Antibody | AffiniPure Donkey Anti-Mouse IgG (H+L) Cy3 (polyclonal) | Jackson ImmunoResearch | CAT#: 715-165-150; RRID: AB_2340813 | 1:400 |
| Antibody | AffiniPure Donkey Anti-Rabbit IgG (H+L) Cy5 (polyclonal) | Jackson ImmunoResearch | CAT#: 711-175-152; RRID: AB_2340607 | 1:400 |
| Antibody | AffiniPure Donkey Anti-Rabbit IgG (H+L) Alexa Flour 647 (polyclonal) | Jackson ImmunoResearch | CAT#: 711-605-152; RRID: AB_2492288 | 1:400 |
| Other (serum) | Normal donkey serum | Jackson ImmunoResearch | RRID: AB_2337258 | N/A |
| Other (DNA stain) | SYTO 13 Green Fluorescent Nucleic Acid Stain | Thermo Fisher Scientific | CAT#: S7575 | 1:2000 |
| Other (Phalloidin stain) | Alexa Fluor 488 Phalloidin | Thermo Fisher Scientific | CAT#: 12379; RRID: AB_2315147 | 1:40 |
| Other (Chemical) | α-terpineol | Sigma-Aldrich | CAT#: 432628 | N/A |
| Other (histology chemical) | Protargol (silver proteinate) | Winthrop Chemical Co. | Product discontinued | N/A |
| Other (histology chemical) | Silver proteinate | Generated in-house Followed recipe described in *Pan et al. (2013)* | | N/A |
| Other (histology chemical) | Potassium dichromate | Sigma-Aldrich | CAT#: 207802 | N/A |
| Other (HPLC purified water) | HPLC Water | Sigma-Aldrich | CAT#: 270733 | N/A |
| Other (histology chemical) | Osmium tetroxide | Electron Microscopy Sciences | CAT#: 19150 | N/A |
| Other (histology chemical) | Siler nitrate | Electron Microscopy Sciences | CAT#: 21050 | N/A |
| Other (histology chemical) | Propylene oxide | Electron Microscopy Sciences | CAT#: 20401 | N/A |
| Other (embedding resin) | Durcupan ACM resin (4-part component kit) | Sigma-Aldrich | CAT#: 44610 | N/A |
| Other (mounting medium) | Permount mounting medium | Fisher Scientific | CAT#: SP15-100 | N/A |
| Software, algorithm | Photoshop CC | Adobe Inc | N/A | N/A |
| Software, algorithm | Fiji | Open-source software *Schindelin et al. (2012)* | | z-project plugin |
| Software, algorithm | Helicon Focus | Helicon Soft | N/A | N/A |
| Other (microscope) | LSM 5 Pascal confocal microscope | Zeiss | N/A | N/A |
| Other (microscope) | Orthoplan light microscope | Leitz | N/A | N/A |

To screen for putative mushroom bodies or centers in corresponding locations we employed anti-DC0 immunostaining of lateral protocerebra. Identified centers were then subjected to various histological techniques to detect character arrangements that define the hexapod-stomatopod mushroom body ground pattern. The nuclear stain Syto13 was used to identify neuronal cell bodies and

to distinguish minute basophilic cell bodies associated with anti-DC0-immunoreactive centers. Antisera were used to reveal neuroarchitectures as well as putative serotoninergic and dopaminergic neurons belonging to mushroom body input or output neurons (MBINs, MBONs; *Aso et al., 2014a*; *Aso et al., 2014b*) and feedforward or recurrent GABAergic elements within mushroom body circuitry (*Perisse et al., 2016*; *Bicker et al., 1985*; *Leitch and Laurent, 1996*). Reduced silver staining and α-tubulin immunohistology was employed to reveal the fibroarchitecture of the lateral protocerebrum. Golgi impregnation methods revealing quantities of single neurons were used to compare neuronal organization in mushroom bodies and their suspected 'hemiellipsoid body' homologues in Reptantia.

## Animals

Animals used in this study were obtained from a variety of commercial vendors. Coral banded cleaner shrimps (*Stenopus hispidus*; n = 25) and fiddler crabs (*Uca minax*; n = 12) were purchased from Salty Bottom Reef Co. (New Port Richey, FL). Land hermit crabs (*Coenobita clypeatus*; n = 13), the crayfish *Orconectes immunis* (n = 8) and *Procambarus clarkii* (n = 10), and cockroaches (*Periplaneta americana*; n = 20) were obtained from Carolina Biological Supply Co. (Burlingham, NC). Pistol shrimps (*Alpheus bellulus*; n = 20) and some *S. hispidus* (n = 3) were purchased from LiveAquaria (Rhinelander, WI). White-legged shrimp *Penaeus vannamei* (n = 24) and stomatopods (*Neogonodactylus oerstedii*; n = 25) were purchased from Gulf Specimen Marine Laboratories (Panacea, FL). Marine crustaceans were shipped directly to the Arizona laboratory aquaria where they were held in marine tanks maintained with filtered water that approximated ocean salinity (approx. 35 ppt). Freshwater crayfish were kept in tanks with filtered RO (reverse osmosis) water. Aquaria were kept at room temperature and maintained on a 12 hr light/dark cycle. Animals were fed a diet of bloodworms and dried shrimp pellets.

Additional specimens were collected and maintained at Friday Harbor Laboratories (University of Washington; Friday Harbor, WA). Spiny lebbeids (*Lebbeus groenlandicus*; n = 37), squat lobsters (*Munida quadrispina*; n = 2), northern crangon (*Crangon franciscorum*; n = 10), Dana's blade shrimp (*Spirontocaris lamellicornis*; n = 3), *Nebalia pugettensis* (n = 4), *Ligia pallasii* (n = 3), and horned shrimp (*Paracrangon echinata*; n = 8) were collected by trawling along the Puget Sound near San Juan Island. Ghost shrimps (*Neotrypaea californiensis*; n = 11), northern hooded shrimp (*Betaeus harrimani*; n = 3), and purple shore crabs (*Hemigrapsus nudus*; n = 40) were collected along shorelines near Friday Harbor, as were hairy hermit crabs (*Pagurus hirsutiusculus*; n = 35). All animals collected at Friday Harbor were kept alive in shaded outdoor tanks with constantly circulating seawater until time of processing.

## Antibodies

A variety of antibodies were used in this study to visualize neural architecture and identify regions of neural connectivity likely involved with learning and memory. A comprehensive list of all antibodies used here, as well as their source, clonality, concentration, and supplier can be found in the Key Resources Lookup Table.

To detect regions of synaptic densities and cytoskeletal elements, antibodies against synapsin and α-tubulin were used, respectively. Immunostaining against synapsin was carried out using a monoclonal antibody raised against *Drosophila* GST-synapsin fusion protein (SYNORF1; *Klagges et al., 1996*). Two different antibodies were used to label the cytoskeletal protein, α-tubulin. The first was raised in mice against the protozoan *Tetrahymena* α-tubulin (*Thazhath et al., 2002*). The second was generated in rabbit against human α-tubulin, and was used to allow co-labelling in experiments in which the other primary antibody was raised in mouse. Antibodies against synapsin and α-tubulin have been widely used as neural markers across distant phyla, suggesting that both proteins have been highly conserved across evolutionary time (*Harzsch et al., 1997*; *Harzsch and Hansson, 2008*; *Stemme et al., 2012*; *Sullivan et al., 2007*; *Brauchle et al., 2009*; *Andrew et al., 2012*). Additional antibodies raised against serotonin (5HT), glutamic acid decarboxylase (GAD), and tyrosine hydroxylase (TH) were used to further aid in the visualization of neural arrangements. Serotonin is a ubiquitous neurotransmitter, and has previously been used across invertebrate phyla as a marker for phylogenetic analyses (*Harzsch and Waloszek, 2000*; *Harzsch, 2004*; *Stemme et al., 2012*; *Dacks et al., 2006*). GAD and TH are enzymatic precursors to

gamma-aminobutyric acid (GABA) and dopamine, respectively, and were used in this study to detect putative GABAergic and dopaminergic neurons. Anti-GAD has previously been used to identify GABAergic neurons in remipedes (*Stemme et al., 2016*) and crustaceans (*Wolff et al., 2017*; *Sayre and Strausfeld, 2019*). Similarly, anti-TH has been used as a marker to detect putative dopaminergic neurons. Co-labelling experiments have revealed that anti-GAD and anti-TH reliably label GABAergic or dopaminergic cells when used in conjunction with primary antibodies against either GABA or dopamine, respectively (*Cournil et al., 1994*; *Crisp et al., 2002*; *Stern, 2009*; *Stemme et al., 2016*).

Lastly, antibodies against DC0 and RII were used to detect regions likely to be involved with learning and memory. These antibodies recognize the major regulatory (RII; *Li et al., 1999*) and catalytic (DC0) subunits of the *Drosophila* c-AMP-dependent enzyme, protein kinase A (PKA; *Lane and Kalderon, 1993*; *Skoulakis et al., 1993*; *Crittenden et al., 1998*). Anti-RII and anti-DC0 appear to label identically, and Western blot analysis has shown that anti-DC0 reliably labels the expected band across remipedes, crustaceans, and insects (*Wolff et al., 2012*; *Stemme et al., 2016*).

## Immunohistochemistry

Animals were anesthetized by cooling on ice. Heads bearing both eyestalks were removed and placed immediately in freshly made ice-cold fixative containing 4% paraformaldehyde (PFA; Electron Microscopy Sciences; Cat# 15710; Hatfield, PA) and 10% sucrose in 0.1 M phosphate-buffered saline (PBS; Sigma Aldrich; Cat# P4417; St. Louis, MO). Eyestalk and midbrain neural tissue were dissected free and allowed to fix overnight at 4°C. The following day, tissue was rinsed twice in PBS and transferred to Eppendorf tubes containing a preheated agarose-gelatin solution consisting of 7% gelatin, 5% agarose. The tissue was left in this mixture and kept warm in a 60°C water bath for 1 hr adapted from *Long (2018)*. Tissue was then transferred to a plastic mold containing fresh embedding medium and allowed to solidify at 4°C for 5–10 min. Next, blocks were removed from the mold, trimmed, and post-fixed in 4% PFA in PBS for 1 hr at 4°C. Blocks were then rinsed twice in PBS and vibratome sectioned at 60 µm (Leica; Nussloch, Germany).

Following sectioning, tissue slices were rinsed twice in PBS containing 0.5% Triton-X (PBST; Electron Microscopy Sciences; Cat# 22140; Hatfield, PA) and blocked in a solution of 5% normal donkey serum (Jackson ImmunoResearch; RRID:AB_2337258; West Grove, PA) for 1 hr. The primary antibody was then added and the sections were left overnight on a gentle shaker at room temperature (RT). TH immunolabelling required an alternative immunostaining method, as is described below. Control experiments in which the primary antibody was not added resulted in the complete absence of immunostaining.

The next day, sections were rinsed 6 times over the course of an hour in PBST. Meanwhile, 2.5 µl of secondary antibody was added to Eppendorf tubes at a concentration of 1:400 and centrifuged at 11,000 *g* for 15 min. Secondary IgG antibodies used in this study were raised in donkey and conjugated to either Cy3, Cy5, or Alexa 647 fluorophores (Jackson ImmunoResearch; RRID: AB_2340813; RRID: AB_2340607; RRID: AB_2492288, respectively; West Grove, PA). After centrifugation, the top 900 µl from the secondary antibody solution was added to the tissue sections, and the sections were left to soak on a gentle shaker overnight at RT. Following secondary antibody treatment, sections were rinsed twice in 0.1M Tris-HCl (Sigma Aldrich; Cat# T1503; St. Louis, MO) buffer. To label cell nuclei, Syto13 (Thermo Fisher Scientific; Cat# S7575; Waltham, MA) was used at a concentration of 1:2000 and left to incubate for 1 hr on a vigorous shaker. Lastly, tissue sections were washed in Tris-HCl for six changes over the course of an hour before being mounted in a medium consisting of 25% glycerol, 25% polyvinyl alcohol, and 50% PBS using #1.5 coverslips (Fisher Scientific; Cat# 12-544E; Hampton, NH).

In some preparations phalloidin conjugated to Alexa 488 (Thermo Fisher Scientific; RRID: AB_2315147; Waltham, MA) was used to label the cytoskeletal protein, F-actin. Sections processed this way were incubated in a solution containing the conjugated phalloidin at a concentration of 1:40 in PBST for 2–3 days following secondary antibody treatment. Sections were left in the dark on a gentle shake at RT for the duration of this time, and then subsequently mounted as described above.

For TH immunolabelling, neural tissue was treated using a much shorter fixation time of just 30–45 min. Longer fixation periods resulted in a reduction or complete loss of immunostaining, as has been previously reported in locusts (*Lange and Chan, 2008*). Additionally, antibody labelling was best preserved when carried out on whole mounts before sectioning, as previously reported in

*Cournil et al. (1994)*. Following fixation, whole mount tissue was rinsed twice in PBS and then twice in PBST over the course of 20 min. Brains were then blocked as described above for 3 hr on a gentle shake. Primary antibody solution was added to the tissue, and the preparations were exposed to microwave treatment. This included two cycles of 2 min low power, 2 min no power, under a vacuum at 18°C. Whole brains were then left for 2–3 days in primary antibody solution on a gentle shake at RT and were subjected to microwave treatment each day of antibody incubation. Following primary antibody treatment, tissue was rinsed 6 times in PBST for 1 hr and left to incubate for 1–2 days in secondary antibody (secondary antibodies were centrifuged as described above). Afterwards, the whole mounts were rinsed 6 times over the course of an hour, embedded, sectioned, and mounted as described above.

## Bodian silver staining

Nerve fibers were resolved using Bodian's reduced-silver stain (*Bodian, 1936*). Nervous tissue was dissected in acetic alcohol formalin fixative (AAF; 10 ml 16% paraformaldehyde, 2.5 ml glacial acetic acid, 42 ml 80% ethanol) and left to fix for 3 hr at RT. Tissue was rinsed twice in 50% ethanol and then transferred through an ethanol series of 70%, 90%, and two changes of 100% ethanol for 10 min each. Next, dehydrated tissue was soaked in α-terpineol (Sigma Aldrich; Cat# 432628; St. Louis, MO) for 15 min before being transferred into xylenes for an additional 15 min. Tissue was then embedded in paraffin, sectioned at 12 μm, and processed following Bodian's original method (*Bodian, 1936*); for further details see *Sayre and Strausfeld (2019)*. It is important to note that one of the most deterministic staining variables in this method is the silver proteinate, more commonly known by its commercial name, Protargol (*Bodian, 1936*; *Pan et al., 2013*). To the dismay of many anatomical biologists, effective forms of Protargol have been commercially discontinued in the past decade. Other forms of silver proteinate that have become commercially available since the disappearance of Protargol have proven to be ineffective for this staining procedure (*Pan et al., 2013*). For the present study, silver proteinate was generated in-house, following the method described in *Pan et al. (2013)* (*Figures 4C*, *5A,D–E*, *6F* and *7C*). The exceptions are *Figures 2D* and *9A*, which used the now unobtainable French 'Roche Proteinate d'Argent', and *Figure 10A*, which was processed using Protargol (Winthrop Chemical Co., New York, NY).

## Golgi impregnations

Animals were anesthetized over ice, and midbrain and eyestalk neural tissue was dissected free from the exoskeleton and enveloping sheath in an ice-cold fixative solution containing 1 part 25% glutaraldehyde (Electron Microscopy Sciences; Cat# 16220; Hatfield, PA) to 5 parts 2.5% potassium dichromate (Sigma Aldrich; Cat# 207802; St. Louis, MO) with 3–12% sucrose in high pressure liquid chromatography (HPLC)-grade distilled water (Sigma Aldrich; Ca#270733). Dissected whole brains were then transferred to fresh fixative and left overnight in a dark cupboard at RT. The rest of the protocol was also carried out at RT, with preparations kept in the dark whenever possible. The use of metal tools was avoided at all stages following the dissection to prevent contamination with the highly reactive silver nitrate.

The next morning, brains were briefly rinsed in 2.5% potassium dichromate and placed in a solution of 2.5% potassium dichromate and 0.01% osmium tetroxide (Electron Microscopy Sciences; Cat# 19150; Hatfield, PA) for around 12 hr. After osmification, neural tissue was post-fixed for 24 hr using freshly made 1:5 admixture of 25% glutaraldehyde:2.5% potassium dichromate, omitting sucrose.

Before silver impregnation, tissue was washed three times for 10 min each in 2.5% potassium dichromate and then transferred to glass containers containing fresh 0.75% silver nitrate (Electron Microscopy Sciences; Cat# 21050; Hatfield, PA) in HPLC water. The brains were quickly transferred to fresh silver nitrate two more times, then left overnight in silver nitrate. The next day, brains were rinsed twice in HPLC water. Some tissue was dehydrated and embedded as described below. For mass impregnation of neurons, brains were treated with a double impregnation in which the osmification step followed by a second treatment of silver nitrate was repeated.

After silver impregnation, brains were rinsed twice in HPLC water and dehydrated through a four step ethanol series from 50% to 100% ethanol at 10 min intervals. Ethanol was then replaced by propylene oxide (Electron Microscopy Sciences; Cat# 20401; Hatfield, PA), and tissue was left for 15

min before being transferred through an increasing concentration of Durcupan embedding medium (Sigma Aldrich; Cat# 44610; St. Louis, MO) in 25%, 50%, and 75% propylene oxide over the course of 3 hr. Tissue was left overnight in 100% liquid Durcupan and then polymerized in BEEM capsules at 60°C for 12–24 hr. Cooled blocks were sectioned at a thickness of 20–40 μm and mounted using Permount mounting medium (Fisher Scientific; Cat# SP15-100; Hampton, NH).

## Imaging

Immunohistochemically stained brain sections were imaged using either a Zeiss Pascal 5 or LSM880 laser scanning confocal microscope (Zeiss; Oberkochen, Germany). Maximum intensity z-projections were produced using FIJI (ImageJ; *Schindelin et al., 2012*); z-project plugin). Brightness, contrast, and color intensity were adjusted using Adobe Photoshop CC (Adobe Systems Inc; San Jose, California). Histological sections were imaged at multiple focal planes (0.5–1 μm increments in the z-direction) using a light microscope (Leitz; Orthoplan), and then stitched together using Helicon Focus (Helicon Soft; Kharkov, Ukraine).

## Terminology

Terms and abbreviations adhere, insofar as possible, to recommendations by the insect brain consortium in its determination of terms applicable to insect and crustacean (Pancrustacea) central nervous systems. See *Ito et al. (2014)*.

## Acknowledgements

The research described here is supported by the National Science Foundation under Grants No. 1754798 awarded to NJS and No. 1754610 to GHW, as well as the University of Arizona's Center for Insect Science, and funding to NJS from the University of Arizona Regents' Fund. Our gratitude is directed to Daniel Kalderon, Columbia University, New York, for supplying the DC0 and RII antibodies, as he has during the last decade and longer. We thank the staff of the University of Washington's Friday Harbor Marine Laboratories, San Juan, for their hospitality and unfailing help in obtaining living specimens. We have much profited from illuminating exchanges with Joanna M Wolfe, Harvard University, Cambridge, MA, for her crucial advice on the use of geological time lines and for identifying a number of errors in our original assessments. Charles Derby, Georgia State University, provided helpful advice on penaeid life cycles and habits. We are indebted to Camilla Strausfeld for critically discussing versions of the final manuscript, suggesting many improvements and meticulously editing the text.

## Additional information

### Funding

| Funder | Grant reference number | Author |
| --- | --- | --- |
| National Science Foundation | 1754798 | Nicholas James Strausfeld |
| National Science Foundation | 1754610 | Gabriella Hanna Wolff |

The funders had no role in study design, data collection and interpretation, or the decision to submit the work for publication.

### Author contributions

Nicholas James Strausfeld, Conceptualization, Resources, Data curation, Formal analysis, Funding acquisition, Investigation, Visualization, Methodology, Writing - original draft, Writing - review and editing; Gabriella Hanna Wolff, Conceptualization, Resources, Formal analysis, Funding acquisition, Validation, Investigation, Visualization, Methodology, Writing - review and editing; Marcel Ethan Sayre, Conceptualization, Resources, Data curation, Formal analysis, Investigation, Visualization, Methodology, Writing - original draft, Writing - review and editing

## Author ORCIDs

Nicholas James Strausfeld (iD) https://orcid.org/0000-0002-1115-1774
Gabriella Hanna Wolff (iD) https://orcid.org/0000-0002-0075-4975
Marcel Ethan Sayre (iD) https://orcid.org/0000-0002-2667-4228

## Decision letter and Author response

Decision letter https://doi.org/10.7554/eLife.52411.sa1
Author response https://doi.org/10.7554/eLife.52411.sa2

## Additional files

### Supplementary files

• Transparent reporting form

### Data availability

Data generated and analyzed for this study derive from histological material kept and curated in the first author's laboratory at the Department Neuroscience, University of Arizona.

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
