## [Decision Letter]

**Acceptance summary:**

The work examines the evolutionary diversity of mushroom body organization across a diverse group of crustaceans, the eumalacostraca, and supports the hypothesis that insect-like mushroom bodies comprise the ground pattern organization of the higher olfactory/multisensory centers across malacostracan crustaceans. The data described further support the unity of Pancrustacea as a subphylum.

**Decision letter after peer review:**

Thank you for submitting your article "Mushroom body homology and divergence across pancrustacea" for consideration by *eLife*. Your article has been reviewed by three peer reviewers, including Catherine Emily Carr as the Reviewing Editor and Reviewer #1, and the evaluation has been overseen by a Senior Editor. The following individuals involved in review of your submission have agreed to reveal their identity: Daniel Osorio (Reviewer #2); Paul Katz (Reviewer #4).

The reviewers have discussed the reviews with one another, and the Reviewing Editor has drafted this decision to help you prepare a revised submission.

Summary:

A previous study from the authors found evidence for homology between the stomatopod hemiellipsoid and insect mushroom bodies. The present submission extends the comparison to a larger range of crustaceans and supports the hypothesis that these dome- or cap-like neuropils typify arthropod lineages, although more variable than those of insects. There was considerable enthusiasm for the outstanding presentation of arthropod anatomy.

Essential revisions:

Concerns were felt to be readily addressable, since they focused on the paper's mixture of results and discussion. Reviewers suggest that the authors separate the Results section from Discussion section to make it clear what the observations are and what the conclusions drawn from those observations are. Further, the reviewers suggest that the authors focus on the interesting question of morphological homology; they felt that some functional conclusions could not be justified by the results.

The manuscript refers to studies by the same group that identify mushroom bodies in other arthropods notably chelicerates (including an Amblipygid), which implies that these structures are present beyond the mandibulates. At the same time the set of characters that are used to define these structures leaves some room for doubt as to which are attributable to convergent evolution and which to common ancestry. Although this type of question is open to pedantry and sterile debate, a clearer introduction to the evidence for homology would be of value, especially for non-specialist readers.

With respect to the functional interpretation of the mushroom bodies, one part of the argument suggests that their primary role, and certainly the reason for their large size in certain groups is for spatial navigation/learning especially in 3-D environments, another part connects the MBs to the olfactory system, and relates their development to olfactory processing and olfactory learning. This can give the impression that the arguments are ad hoc. Since the main contribution here is to greatly increase our understanding of the evolutionary/adaptive variation in the mushroom bodies it would be helpful to explain more explicitly the premises of the arguments proposed to account for the variation, for example by providing summary figure (companion to the current Figure 16) outlining some key conclusions/hypotheses about the functional roles of the different components of the mushroom bodies that might account for their enlargement or reduction in various mandibulate/crustacean+ hexapod taxa.

[Editors' note: further revisions were suggested prior to acceptance, as described below.]

Thank you for submitting your revised article "Mushroom Body Evolution Demonstrates Homology and Divergence across Pancrustacea" for consideration by *eLife*. Your revised article has been reviewed by a Reviewing Editor and Timothy Behrens as the Senior Editor.

Summary:

A previous study from the authors found evidence for homology between the stomatopod hemiellipsoid and insect mushroom bodies. The present submission extends the comparison to a larger range of crustaceans and supports the hypothesis that these dome- or cap-like neuropils typify arthropod lineages, although more variable than those of insects. There was considerable enthusiasm for the outstanding presentation of arthropod anatomy.

Essential revisions:

Please rewrite paragraph one of the Discussion section to improve clarity. The discussion itself is long, given that the results already include background material. Please shorten where possible.

Add citations as needed to the statements in subsection “Olfactory organization and mushroom body divergence”.

Consider rewriting subsection “Mushroom bodies unify Mandibulata (Pancrustacea and Myriapoda” to be less disparaging. It is understood that a goal of this paper is to show how the transformation from a columnar lobed mushroom body morphotype to a calycal morphotype can be followed across eumalacostracan evolution, but this part of the discussion section could be shortened.

[Editors' note: further revisions were suggested prior to acceptance, as described below.]

Thank you for submitting your revised article "Mushroom Body Evolution Demonstrates Homology and Divergence across Pancrustacea" for consideration by *eLife*. Your article has been reviewed by a Reviewing Editor.

Your Discussion section remains very long, so please consider shortening it. With respect to the discussion about the Machon paper, I have now read that paper, and it seems to me that they agree with you. They write:

"Recent evidence suggests that, despite many morphological differences, these protocerebral structures of insects and crustaceans nevertheless share common architectural, physiological and neurochemical features suggesting a homology of their very basic neuronal circuitry (Brown and Wolff, 2012; Maza et al., 2016; Wolff et al., 2012; Wolff et al., 2017; Wolff and Strausfeld, 2015."

Nevertheless, your Discussion section reads "Yet, in the ensuing years the belief that the domed hemiellipsoid bodies of crustaceans are fundamentally different structures from the lobed mushroom bodies of insects has become not only widely accepted, but specifically advocated as a basis for the ground pattern of the malacostracan brain (Sandeman et al., 2014; Wittfoth et al., 2019; Machon et al., 2019; Krieger et al., 2019). […] Considering mushroom bodies exclusively as hemiellipsoid morphologies is problematic for reaching consensus about pancrustacean brain evolution (compare the present Figure 12 with Machon et al., 2019; loc. cit. Figure 12)."

I cannot see this point in your Figure 12 and the Machon Figure 12. Please modify your Discussion section so that the key theoretical points are clearer, or if you wish, we can send the paper out to the reviewers.

---

## [Author Response]

Essential revisions:Concerns were felt to be readily addressable, since they focused on the paper's mixture of results and discussion. Reviewers suggest that the authors separate the Results section from Discussion section to make it clear what the observations are and what the conclusions drawn from those observations are. Further, the reviewers suggest that the authors focus on the interesting question of morphological homology; they felt that some functional conclusions could not be justified by the results.

Those elements of the original manuscript that have the character of a discussion have either been deleted or integrated into the Discussion section. The first part of the Discussion section has, accordingly, been extensively revised.

It is not so much that discussions were intermixed with results, but that introducing each taxon needs to be separated from the neuroanatomical description of that taxon. Many published neuroanatomical descriptions make the mistake in assuming that the reader will know everything that the authors know. Background information brings relevance to the species under consideration. In this revision, we have divided the presentation of each taxon into “Background” and “Observations.” The first provides a condensed overview of the taxon’s evolutionary status, habitat, its habits insofar that these are known, and other information. “Observations” provide the data as neuroanatomical descriptions.

The manuscript refers to studies by the same group that identify mushroom bodies in other arthropods notably chelicerates (including an Amblipygid), which implies that these structures are present beyond the mandibulates. At the same time the set of characters that are used to define these structures leaves some room for doubt as to which are attributable to convergent evolution and which to common ancestry. Although this type of question is open to pedantry and sterile debate a clearer Introduction to the evidence for homology would be of value, especially for non-specialist readers.

We originally referred to the expanded anti-DC0 domains in Amblypygi to suggest that a similar expanded domain in shore crabs could have evolved from a mushroom body like ancestry. We agree that this was an insufficient comparison on its own without describing any evidence supporting the amblypygid domain as derived from a mushroom body equipped with a columnar lobe. Those are present in more basal arachnids, as described in an earlier account (Wolff and Strausfeld, 2015). The present revision now expands this discussion to refer to observations showing that evolutionary basal arachnids possess the “standard” mushroom body ground pattern, as defined by its columnar lobe, parallel fibers and organization of input and output pathways. Current knowledge of the chelicerate molecular phylogeny (Giribet, 2018; Sharma et al., 2014), indicates that large folded anti-DC0 positive domains we observe in Amblypygi, a derived lineage, have evolved from the ancestral standard mushroom body morphology and that lineages even younger than Amblypygi have reverted to the “standard” morphology. As now explained in the revised Discussion section, chelicerates support the suggestion that an expanded anti- DC0 domain in Brachyura may similarly have originated from the “standard” mushroom body ground pattern observed in stomatopods and certain caridean lineages.

With respect to the functional interpretation of the mushroom bodies, one part of the argument suggests that their primary role, and certainly the reason for their large size in certain groups is for spatial navigation/learning especially in 3-D environments, another part connects the MBs to the olfactory system, and relates their development to olfactory processing and olfactory learning. This can give the impression that the arguments are ad hoc. Since the main contribution here is to greatly increase our understanding of the evolutionary/adaptive variation in the mushroom bodies it would be helpful to explain more explicitly the premises of the arguments proposed to account for the variation, for example by providing summary figure (companion to the current Figure 16) outlining some key conclusions/hypotheses about the functional roles of the different components of the mushroom bodies that might account for their enlargement or reduction in various mandibulate/crustacean+ hexapod taxa.

Regarding the suggestion that we discuss the functional components of the mushroom body, and what accounts for their differences across lineages: our discussion does this on a broader scale than focusing on specific anatomical details, for which there is, as yet, a paucity of functional information. Our arguments are not ad hoc, however: they address the overarching observation that whereas mushroom bodies are highly conserved across hexapod lineages, this is not the case for crustaceans where there is overwhelming evidence for highly divergent evolution of the mushroom body ground pattern. To better bring this stark difference to the fore, we have now reorganized the relevant Discussion sections to first address the possible evolutionary consequences, relating to mushroom body diversification, of major differences of the olfactory pathways between insects and crustaceans. Our proposition is that genetic hard-wiring (labeled-line organization) of the hexapod olfactory system accompanies the highly constrained diversification of the insect mushroom body and could be the proximate cause of such constraint. We hypothesize that the all-to-all connectivity of the crustacean olfactory pathway, which is distinct from that of the insect in lacking any recognizable labelled line organization, may have permitted highly relaxed divergent evolution of mushroom bodies in crustaceans. As far as we are aware, this suggestion, which is novel, identifies the cardinal distinction between the considerable diversification of the mushroom body ground pattern across crustaceans, but not across hexapods.

Those considerations are followed by a discussion about the diversification of the mushroom body ground pattern across pancrustacean lineages. Based on what is known (which is still sparse) about the ecologies and behaviors of the investigated species we propose that the prominence of a mushroom body’s columnar lobe appears to reflect a species’ ability to negotiating complex three-dimensional ecologies. Stomatopod and caridean lineages, and one group of reptantians, exploiting elaborate three-dimensional ecologies have retained the ancestral columnar elements of their mushroom bodies. So have dicondylic insects, which essentially operate under the same constraints of a three-dimensional world, albeit not in an aquatic but in an aerial or subaerial context.

[Editors' note: further revisions were suggested prior to acceptance, as described below.]

Essential revisions:

Please rewrite of paragraph one of the Discussion section to improve clarity. The Discussion section itself is long, given that the results already include background material. Please shorten where possible.Add citations as needed to the statements in subsection “Olfactory organization and mushroom body divergence”.Consider rewriting subsection “subsection “Mushroom bodies unify Mandibulata (Pancrustacea and Myriapod” to be less disparaging. It is understood that a goal of this paper is to show how the transformation from a columnar lobed mushroom body morphotype to a calycal morphotype can be followed across eumalacostracan evolution, but this part of the Discussion section could be shortened.

We thank the reviewing editor and senior editor for their further letter and suggestions regarding the revised manuscript.

Regarding the suggestion to shorten the Discussion section: although the first part of the discussion was rewritten for clarity in the first revision, the discussion was in its totality the same length as in the original manuscript although its sequences were rearranged. As noted in our response accompanying the first revision, background information in the Results section is still retained in situ and is not repeated in the Discussion section. We also followed the request in the first review to expand the discussion regarding comparisons with Arachnida. In this present revision, we have deleted the introductory paragraph of the Discussion section. Further reductions, indicated as strike outs in the marked-up copy, account for the removal of, in total, 289 words.

The editors are correct regarding the described evolutionary trends; albeit the evolved transition of the morphotype, from columnar to calycal, is more nuanced, as we demonstrate for Anomura, Brachycera, and total Insecta. But regarding the point of view we express in subsection “Mushroom bodies unify Mandibulata (Pancrustacea and Myriapod”, we think it is incorrect to dismiss any such transition, which is the position taken in a recent publication carrying considerable weight. Therein is stated that, for “simplicity,” discussing columnar attributes will not considered. As a consequence, the columns of mushroom bodies are airbrushed from existence, as demonstrated by Figure 12 in Machon et al., 2019. At its most uncharitable, this could be seen as a distortion of evidence. Our opinion is that such a published position is not a throw-away aside but one that effectively shuts down further discussion. That action should be as contestable as any incorrect data set.

The editors consider that the manner in which we originally contest this could be seen as disparaging to the authors that promulgate such a view. We agree that one should avoid any ad hominem taint, and we have thought long and hard about how to address this. How, for example, should a biologist studying the evolution of the crown group Reptilia react to the injunction by a morphologist working on Serpentes that “any consideration of legs as typifying the ancestral state of Squamata will be excluded for simplicity.” In this present revision we have edited the relevant text: we do not disparage any person but we do indicate there is a considerable problem when denying a crucial feature of the centres under discussion.

Regarding “citations are needed to the statements in subsection “Olfactory organization and mushroom body divergence””. We have modified this passage accordingly.

[Editors' note: further revisions were suggested prior to acceptance, as described below.]

Your discussion remains very long, so please consider shortening it. With respect to the discussion about the Machon paper, I have now read that paper, and it seems to me that they agree with you. They write:"Recent evidence suggests that, despite many morphological differences, these protocerebral structures of insects and crustaceans nevertheless share common architectural, physiological and neurochemical features suggesting a homology of their very basic neuronal circuitry (Brown and Wolff, 2012; Maza et al., 2016; Wolff et al., 2012; Wolff et al., 2017; Wolff and Strausfeld, 2015."Nevertheless, your discussion reads: "Yet, in the ensuing years the belief that the domed hemiellipsoid bodies of crustaceans are fundamentally different structures from the lobed mushroom bodies of insects has become not only widely accepted, but specifically advocated as a basis for the ground pattern of the malacostracan brain (Sandeman et al., 2014; Wittfoth et al., 2019; Machon et al., 2019; Krieger et al., 2019). […] Considering mushroom bodies exclusively as hemiellipsoid morphologies is problematic for reaching consensus about pancrustacean brain evolution (compare the present Figure 12 with Machon et al., 2019; loc. cit. Figure 12)."I cannot see this point in your Figure 12 and the Machon Figure 12. Please modify your Discussion section so that the key theoretical points are clearer, or if you wish, we can send the paper out to the reviewers.

Thank you for your latest comments. I apologise for a mistake I made in our last response to you, where I proposed the editors compare our Figure 12 and Figure 12 of Machon et al., because it isn’t our Figure 12 that reveals the crucial differences, but our Figure 16. Thus, a typo that by repetition seems to have caused a lot of trouble for everyone.

In any event, we have further shortened the discussion, including the passages referring to Machon et al. However, we also have to respond to your highlighted passage from the Machon et al. We are very aware of this passage. But if one is to take Machon et al.’s Figure 12 seriously, it does not deflect from those authors promoting an apomorphic ancestral ground pattern across Crustacea. If one compares their Figure 12 with our Figure 16, the discrepancy between what is their established view and the results of our present evolutionary study is unambiguous. The passage quoted, where the authors refer to homologies, reveals a contradiction that cannot be ignored. Namely, the homologies we have shown in our works (they cite these as “recent evidence”) refer to the defining characters of the *columnar lobes* of the mushroom body morphotype. Yet, Machon et al., (2019) are explicit in dismissing any reference to columns. “The additional stalked neuropils in the lateral protocerebrum of *N. oerstedii* (Wolff et al., 2017) will not be discussed here for simplicity.” That statement does not simply pertain to the mantis shrimp: their position is transparent in Figure 12, which shows no lobed morphotypes nor mentions in its legend that there might be any. If the lobeless morphotype is the ancestral malacostracan center then how can it possess mushroom body-type homologies ("recent evidence" of circuits, neurochemicals) that are features of the *lobed* mushroom body morphotype. One can’t have it both ways: insisting on lobeless morphotypes as the ground pattern but claiming basic homologies by citing (our) studies that focused on an entirely different entity: the columnar organization of the lobed morphotype.

We hope that this clarifies for you the kind of discrepancies, and evidential differences, that required clarified. We have not, in our manuscript, discussed these as narrated above. To have done so would have taken up more space; and, in any event, the reader will come to her or his own conclusions. In this third revision we have tried to navigate around this problem gingerly, in a manner that shouldn’t cause anyone offense while still presenting and discussing highly divergent views. Correcting one of these is crucial for interpreting evidence that supports the evolution of this learning center: not only across Pancrustacea, but across Panarthropoda and representatives of Spiralia. We trust that the present revision and further text reductions are now acceptable.